# Seed encrusting with salicylic acid: A novel approach to improve establishment of grass species in ecological restoration

**Simone Pedrini** [1]*, **Jason C. Stevens** [2,3], **Kingsley W. Dixon** [1,3]

**1** ARC Centre for Mine Site Restoration, School of Molecular and Life Science, Curtin University, Bentley, Western Australia, Australia, **2** Department of Biodiversity, Kings Park Science, Conservation and Attractions, Kings Park, Western Australia, Australia, **3** School of Biological Sciences, The University of Western Australia, Crawley, Western Australia, Australia

* simone.pedrini@curtin.edu.au

**Data Availability Statement:** All relevant data are within the paper and its Supporting Information files.

## Abstract

To achieve global ambitions in large scale ecological restoration, there is a need for approaches that improve the efficiency of seed-based interventions, particularly in overcoming the bottleneck in the transition from germination to seedling establishment. In this study, we tested a novel seed-based application of the plant stress modulator compound salicylic acid as a means to reduce seedling losses in the seed-to-seedling phase. Seed coating technology (encrusting) was developed as a precursor for optimising field sowing for three grass species commonly used in restoration programs, *Austrostipa scabra*, *Microlaena stipoides*, and *Rytidosperma geniculatum*. Salicylic acid (SA, 0.1mM) was delivered to seeds via imbibition and seed encrusting. The effects of SA on seed germination were examined under controlled water-limited conditions (drought resilience) in laboratory setting and on seed germination, seedling emergence, seedling growth and plant survival in field conditions. Salicylic acid did not impact germination under water stress in controlled laboratory conditions and did not affect seedling emergence in the field. However, seedling survival and growth were improved in plants grown from SA treated seeds (imbibed and encrusted) under field conditions. When SA delivery methods of imbibing and coating were compared, there was no significant difference in survival and growth, showing that seed coating has potential to deliver SA. Effect of intraspecific competition as a result of seedling density was also considered. Seedling survival over the dry summer season was more than double at low seedling density (40 plants/m²) compared to high seedling density (380 plants/m²). Overall, adjustment of seeding rate according to expected emergence combined with the use of salicylic acid via coating could improve seed use efficiency in seed-based restoration.

## Introduction

Almost two-thirds of the world's ecosystems are considered degraded or damaged [1]. Such degradation poses a serious risk to biodiversity and greatly impacts human communities that rely on ecosystem services for their sustenance and wellbeing [2,3]. In cases of extreme

**Funding:** This study was supported by Curtin University in the form of an International Postgraduate Research Scholarship awarded to SP and the Australian Government through the Australian Research Council (ARC) Industrial Transformation Training Centre for Mine Site Restoration in the form of equipment and materials to conduct the study, the publication cost, and a PhD scholarship awarded to SP (Project Number ICI150100041). The views expressed herein are those of the authors and are not necessarily those of the Australian Government or Australian Research Council. The funders had no role in study design, data collection and analysis, decision to publish, or preparation of the manuscript.

**Competing interests:** The authors have read the journal's policy and have the following competing interests: Funding and material support for this research were provided by the Australian Government through the Australian Research Council Industrial Transformation Training Centre for Mine Site Restoration, a consortium of 9 organisations: 2 universities (Curtin, UWA) and 7 industry partners (DBCA/Kings Park, Seraustralasia, BHP, Karara, SMC, Mineral resources, Hanson). This does not alter our adherence to PLOS ONE policies on sharing data and materials. There are no patents, products in development or marketed products associated with this research to declare.

disturbance, as in post mining landscapes, where spontaneous regeneration may not be feasible or effective, restorative interventions to assist in returning the functionality, diversity, and structure of degraded landscapes are required [4,5]. Native seeds of appropriate-local origin is the most widespread method for undertaking restoration when the land or, as in the case of mining, topsoil, has limited natural regenerative capacity [6,7]. However, abiotic factors such as nutrient-impoverishment, chemically and physically hostile soil conditions [8] and low or unpredictable water availability [9], combined with biotic variables such as seed predation [10] and competition with exotic species [11], can limit the success of traditional seed-based restoration.

Generally, less than 10% of sown native seeds become established plants, with significant bottlenecks occurring at the seedling emergence phase [12] and survival through the first summer drought [13]. Given the high cost and often highly limited availability of native seed [14], improving the efficiency of seed use (deployment, germination, and plant establishment) is crucial if ecological restoration is to be delivered at the expected landscape scale [15]. To address issues related to logistical constraints on seed delivery and seedling establishment, the crop seed industry has developed technologies, such as seed coating, that could be adapted and applied to native seed in restoration programs [16].

Seed coating is the practice of covering seeds with external materials, sometimes including active ingredients, conferring protection and improved physiological performance to seeds [17]. Seed coating has been tested on native seeds in different restoration scenarios to overcome specific limitations such as water repellency [18], soil crusting [19], and seed predation [20]. However, despite promising results of seed coating leading to improved seedling emergence, few studies have so far attempted to improve native seed germination and seedling resistance to abiotic stresses that are major impactors on restoration success [21].

Improving resistance to some abiotic stresses could be conferred by exposure of seeds to salicylic acid (SA), a plant hormone synthesised by many plant species [22]. Salicylic acid is involved in plant growth, developmental regulation [23], signalling [24], thermogenesis and stress response mediation, either by providing resistance or triggering apoptosis [25]. Exogenous application of SA through irrigation, foliar spray, or seed imbibition has increased plant resistance and survival to a wide range of abiotic and biotic stresses [26]. Salicylic acid efficacy in conferring stress resistance is a function of its concentration, with low doses failing to deliver resistance and higher concentrations decreasing resistance by activating cell death pathways [27,28]. The effect of SA on seed germination remains unclear; studies using seeds of crop species have reported improved germination for *Arabidopsis thaliana* under salinity stress [29] and for wheat (*Triticum aestivum*) under drought stress [30], while no effect has been reported for maize (*Zea mays*) [31] or barley (*Hordeum vulgare*) [32]. When tested on seeds of native Australian pasture species under drought and hyper salinity conditions, germination in some species was enhanced by the application of SA [33–35]. Delivery of SA via seed coating has shown some promising results when tested on tobacco seeds, improving germination and seedling growth under drought stress [36], and in corn, inducing resistance to chilling [37].

However, the application of SA via seed coating has never been tested on native species for ecological restoration.

The goal of this study was to evaluate the effects of SA on seed germination success, seedling emergence, survival, and growth on three grass species native to southern temperate Australia. Moreover, by comparing SA delivery methods of imbibition and coating, we tested the efficiency and viability of seed coating technology in delivering the benefits of SA to native grass seeds. In particular, the following hypotheses were tested: 1) the process of seed coating and imbibition, without the inclusion of SA, will not have deleterious impact on seed germination success in laboratory trials or seedling emergence in the field, 2) SA will improve germination

under conditions of water stress and enhance seed germination and seedling emergence in the field, and 3) plant survival and growth in the field will be improved for plants established from SA treated seeds.

## Material and methods

### Species selection and seed processing

Three species of grasses native to temperate and Mediterranean regions of southern Australia —*Austrostipa scabra* (Lindl.) S.W.L. Jacobs & J.Everett, *Microlaena stipoides* (Labill.) R.Br. var. Griffin and *Rytidosperma geniculatum* (J.M.Black) Connor & Edgar var. Oxley (all Poaceae)— were selected on the basis of their predominance in revegetation and restoration activities and for their potential use for pasture in replacement of, mostly European, fodder species. [38]. Seeds were sourced from a commercial provider (Native Seed Pty Ltd, Cheltenham, Victoria) in 2016. To reduce the potential for viability loss, seeds were stored in paper bags on open shelving in a controlled environment (15˚C, and 15% relative humidity, RH) for one year prior to experimentation [39]. Seeds were moved to ambient condition (20–25˚C and 40–50% RH) two weeks prior to experimentation to avoid potential seed damage during the cleaning and encrusting process [40].

Caryopses of each species were extracted from the husk to allow for more homogeneous encrusting and imbibition treatment. Removal of the palea and lemma was performed for each species using sulphuric acid digestion *sensu* Stevens *et al*. 2015 [41], with complete immersion of the floret in a 50% sulphuric acid solution (ACS reagent grade $H_2SO_4$, Sigma-Aldrich, St Louis, USA) for an optimal interval allowing for the weakening of floret structures without reducing germination potential. Immersion time for all three species was determined by Pedrini *et al*. 2018 [42], and thus immersion intervals were 90 min for *A. scabra*, 60 min for *M. stipoides* and 20 min for *R. geniculatum*. Acid immersion was followed by neutralisation in 8.4 g $L^{-1}$ sodium bicarbonate ($NaHCO_3$, Sigma-Aldrich, St Louis, USA) solution for 5 min, before rinsing under tap water for 2 min and drying in a Food Lab™ Electronic Dehydrator at 35˚C (Sunbeam, Sydney, Australia) for 3 h. After drying, caryopsis extraction was achieved by gentle rubbing with a rubber mat and sequential sieving and zig-zag air flow separation (Selecta Machinefabriek BV, Enkhuizen, Netherlands).

### Seed treatments

After cleaning, caryopses (hereafter referred to as 'seeds') of each species were subjected to seed imbibition or coating treatment with or without salicylic acid application. The coating treatment used in this experiment is defined as encrusting, because the size and weight of the seed were increased but the shape of the seed remained evident [21] (Fig 1). The treatments tested in this study were: 1) CTRL, untreated seed for control; 2) IN, imbibed seeds without SA; 3) IS, imbibed seeds with SA, 4) EN, encrusted seeds without SA; and 5) ES, encrusted with SA.

Salicylic acid was provided at a concentration of 0.1 mM, a concentration previously shown to be sufficient in confering stress resistance to various species using different delivery methods [28,43,44]. SA solution was prepared by dissolving crystalline SA (Sigma Aldrich, St. Louis, USA) in deionized water for imbibition, and in a 2% Hydroxyethyl cellulose (cellosize QP 09-L, DOW chemicals) solution for encrusting (mixed with a magnetic stirrer for 30 min at 50˚C). For imbibition treatments, seeds were soaked in either SA solution (IS) or deionized water (IN) for 24 h at 20˚C, following previously described methodology for SA delivery to seeds [30,45].

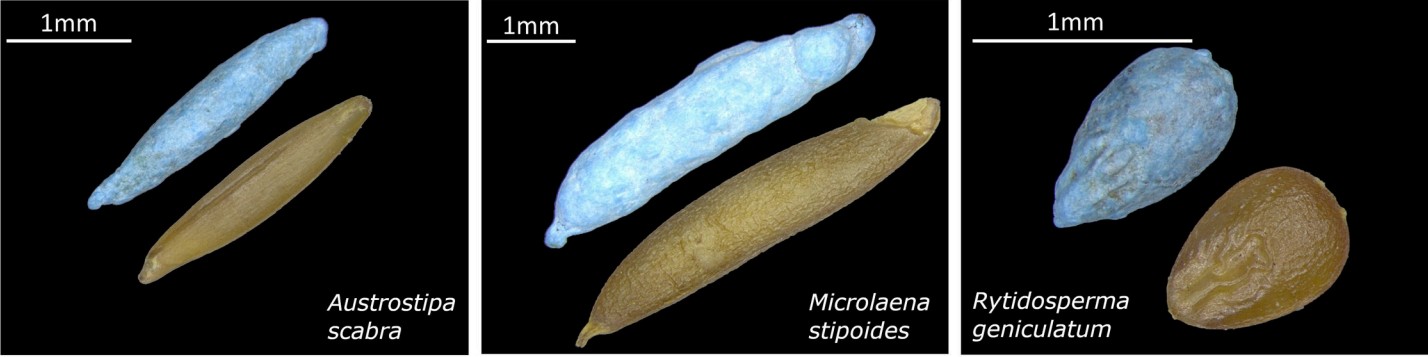

**Fig 1. Seeds of the three grass species tested.** Each image shows encrusted (blue) and untreated-imbibed seeds. Scale bars indicate seed sizes.

Seed encrusting was performed in a 15 cm RRC 150 Lab Coater (Centor Thai, Bangkok, Thailand), *sensu* Pedrini *et al*. 2018 [42]. Liquids were delivered through a compressed air-propelled 0.7 mm airbrush (Ozito tools, Australia). Talc was used as the filler material. Cleaned seeds (10 g) were placed inside the rotary coater, with rotor speed set at 300 RPM, and seeds were initially exposed to liquid spray—with SA (ES) or without (EN)—until moist, before powder was dusted onto the rotating seed mass using a paint brush. By gently tapping the brush on the drum, the powder was slowly released on the rotating mass of seed without affecting their flow.

Wetting and dusting were repeated until 20 g of powder was used and a total of 15 ml of liquid was applied. Seeds were routinely checked to visually ensure even coat coverage and singulation (i.e. each coated unit contains one seed). Following imbibition and encrusting treatments, seeds were placed on trays and dried for 3 h in a Food Lab™ Electronic Dehydrator at 35˚C (Sunbeam, Sydney, Australia).

## Experimental design

The three hypotheses were tested with four experiments: 1) laboratory germination experiment, 2) field germination bag experiment, 3) field line experiment, and 4) field plot experiment. Each hypothesis was tested by using data of multiple experiments as shown in Table 1.

The first experiment was performed in the seed laboratory at the Biodiversity Conservation Centre in Kings Park, Perth, Western Australia. Field experiments (2,3 and 4) were performed at a farming site east of the town of Waroona in Western Australia (32˚ 74' 27" S, 116˚ 00' 36" E, 201 m elevation). The site falls within the native range of all three tested species and offers climatic conditions similar to those of mining operations active in the area and likely to require these species in seed-based rehabilitation following mine closure as well as local farming communities with an interest in native pasture species. The field trial area (6.5 x 18 m) was

**Table 1. Combination of hypotheses and experiments.**

|  | **1) Laboratory germination experiment** | **2) Field germination bag experiment** | **3) Field line experiment** | **4) Field plot experiment** |
|---|---|---|---|---|
| **Hypothesis 1: Seed treatment process effect on germination and emergence** | CTRL vs IN vs EN germination at 0.0 MPa |  | CTRL vs IN vs EN emergence |  |
| **Hypothesis 2: SA effect on germination and emergence** | All treatments germination at all water potentials | All treatments germination | All treatments emergence |  |
| **Hypothesis 3: SA effect on field survival and growth** |  |  | All treatments survival | All treatments survival and growth |

enclosed by a fence to avoid grazing from native marsupials and rabbits (S1 Appendix, Field Experimental Layout in S1 File).

**Laboratory germination experiment.**   Germination tests were performed in Petri dishes lined with two filter papers moistened with 14 ml water or Polyethylene Glycol (PEG) solution and placed in sealed plastic bags to reduce filter paper desiccation. Each week, 2 ml of water or PEG solution was added to ensure suffcent hydration of the filter paper.

In order to test whether SA improved germination success under water-limited conditions, PEG 8000 (Sigma-Aldrich, St Louis, USA) diluted in deionised water at 0, 24.72, 30.78, and 35.90 g/l was used to obtain solutions of 0.0, -0.6, -0.9, and -1.2 MPa water potential at 20°C. The values at reduced water potential resembles the range of water availability recorded in the field during the winter months. (S2 Appendix, Soil Water retention curve in S2 File). Germination tests were performed on four replicates of 25 seeds for each of the five seed treatments (CNTR, IN, IS, EN, ES). Petri dishes were placed in a Biosyn incubator 6000 OP (Contherm, Korokoro, New Zealand) at 20°C with a 12/12 h light/dark photoperiod.

Germination was scored daily for the first 5 days and then at 7, 10 and 15 days, respectively. On day 21, final germination was scored and remaining seeds were examined via cut test to assess viability. Non-viable (i.e. mouldy) seeds were excluded from the total.

**Field germination bag experiment.**   Field seed germination was tested by placing 50 seeds per treatment (CNTR, IN, IS, EN, ES) in 5 cm² sealed nylon mesh bags, over an area of 2 m², and buried on site at a 1cm depth. The bags were collected three weeks after sowing and germination was recorded by counting the seeds with protruding radicles. Each treatment had four replicate bags arranged in a randomised complete block design of four blocks for 15 treatments (five treatments × three species).

**Field line experiment.**   Seedling emergence and survival was tested by sowing 100 seeds per treatment (CNTR, IN, IS, EN, ES) along a meter-long line. Seed were sown at depths of 0.2–0.5 cm, achieved by broadcasting dry soil on top of freshly sown lines. Seedling emergence was scored after 1, 2, 3, 4 6, 8 and 10 weeks. All emerged seedlings were then left to grow to maturity. Plant survival was recorded 45 weeks after sowing. Each treatment had four replicate lines arranged in a randomised complete block design of four blocks for 15 treatments (five treatments × three species).

**Field plot experiment.**   To evaluate plant survival as well as growth, 100 seeds per treatment (CNTR, IN, IS, EN, ES) were manually broadcasted on 0.5 x 0.5 m plots. Seed were sown at depths of 0.2–0.5 cm, achieved by broadcasting dry soil on top of freshly sown plots. A month after sowing, the plots were thinned to 10 randomly selected seedlings, with at least 5 cm between seedlings, resulting in a density of 40 plant/m². Thinning was performed to limit potential intraspecific competition that might have altered plant growth. The selected seedlings were marked with a pin to avoid confusion with other seedlings that could have emerged at a later stage. Forty-five weeks after sowing the surviving plants were counted and harvested, and their height, wet weight, and dry weight recorded. Each treatment had four replicate plots arranged in a randomised complete block design of four blocks for 15 treatments (five treatments × three species).

All experiments were established at the commencement of the growing season in May 2017. Soil temperature and volumetric moisture content (m³/m³) were recorded for the duration of the germination and emergence experiment (10 weeks) with HOBO Micro Station Data Loggers (Onset Computer Corporation, Bourne, MA, USA). The probes were buried at 1 cm. For the 35 weeks following the end of the emergence experiment (July 2017 –March 2018), minimum and maximum temperature and precipitation data were obtained from the Dwellingup weather station, 10 km from the site [46] (Fig 2).

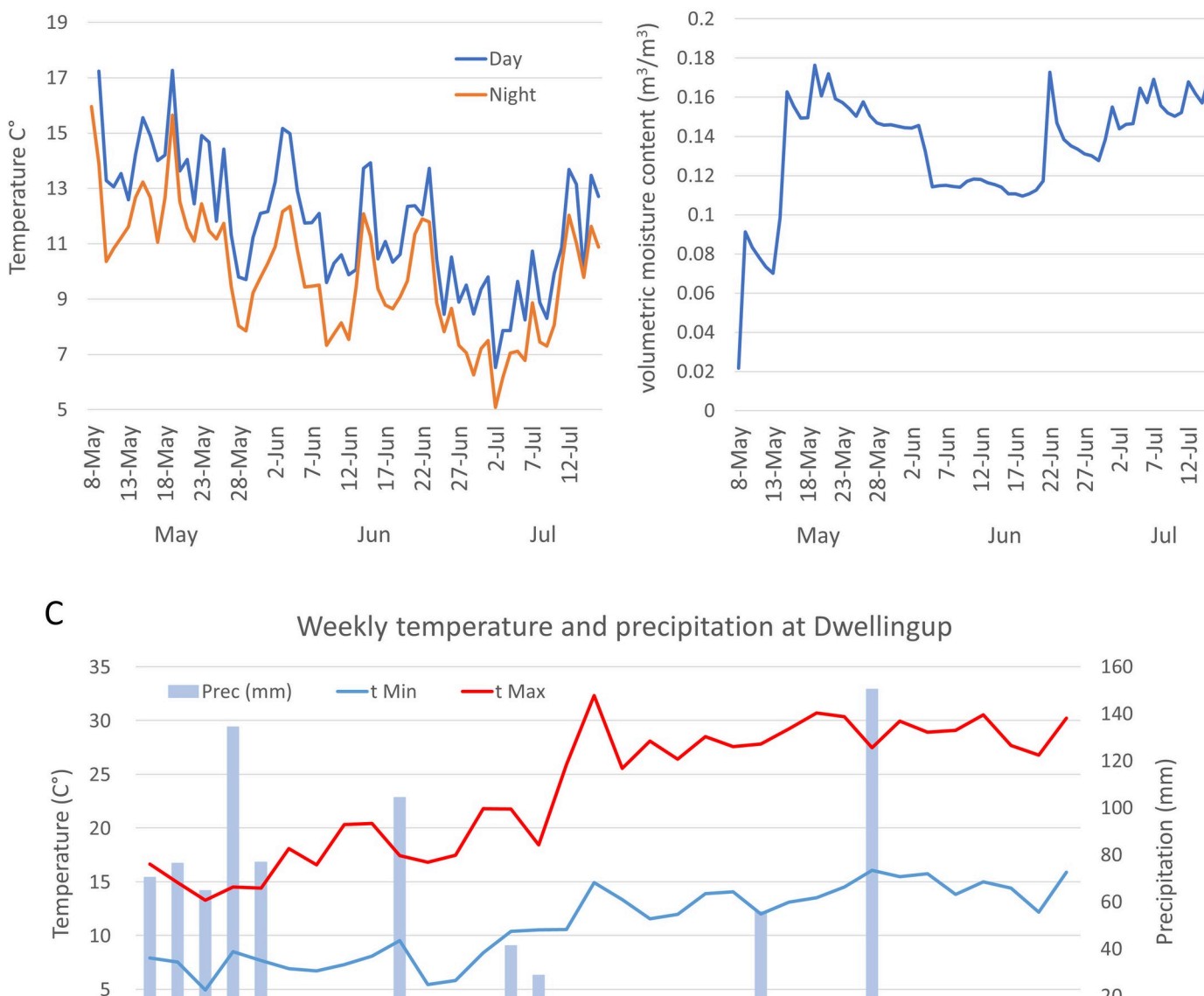

**Fig 2. Climate condition at the field site.** (A) the daily average for day (orange) and night (blue) temperature (B) volumetric water content in the soil at 1 cm depth for the first 10 weeks of the experiment, when germination and emergence were recorded. (C) Weekly maximum (tMax) and minimum (tMin) temperature, and total precipitation (Prec (mm)) for the period between the end of the emergence experiment and the recording of plant survival (July 2017–March 2018) at a nearby meteorological station.

## Statistical analysis

To assess laboratory germination and seedling emergence in the field, non-linear regression models were fitted with the function "drm" of the "DRC" package [9,47,48]. A three parameter

log-logistic model was used:

$$f(x) = \frac{gmax}{1 + \left(\frac{x}{T50}\right)^b}$$

where: b is slope curvature, gmax is final germination and T50 is germination speed, intended as time (days/weeks) required to reach half of the final germination. Parameter comparison on final germination and germination speed were then performed to assess differences among treatments (significance P <0.05). The same analysis was performed on seedling emergence data.

To test the hypothesis of SA effect on germination in the field (in germination bag experiment) and plant survival (in line and in plot experiments), an exact binomial test on the probability of success in a Bernoulli trial was performed between each treatment (confidence level = 0.95).

Plant height and biomass data were fitted in a Linear Mixed-Effects Model using the "lmer" function in the lme4 package in R [49]. Treatment process—comparing control (CTRL) vs. imbibed (IN+IS) vs. encrusted (EN+ES)—and treatment compound—comparing control (CTRL) vs. treated (IN+EN) vs. treated with SA (IS+ES)—were set as fixed variables and replicates were set as a random variable.

ANOVA (Type II Wald chi square tests) was employed to detect significant treatment effects. If such significance was detected, a pairwise t-test was performed to compare the levels within the treatment. All data analysis was performed in the R statistical environment [50].

## Results

### Effect of encrusting vs. imbibition treatment on seeds

Encrusting (EN) had higher or similar germination than the control (CTRL), whilst imbibition (IN) at times resulted in lower germination than the CTRL (Fig 3). Final germination of *A. scabra* treated seed (EN+IN), tested in lab conditions, was not different from that of the CTRL, and only slightly but significantly (P < 0.001) increased in germination speed (T50) of 0.5 days, for both IN and EN seed. When tested in field conditions, EN had lower final emergence than the CTRL (CTRL: 52 ± 1.6%, EN: 45 ± 2.4%, P < 0.001) while IN seeds showed no difference compared to the CTRL.

Under laboratory conditions, *M. stipoides* seeds that underwent EN treatment had higher germination (86 ± 2.1%) than the CTRL (73 ± 2.2%, P < 0.001): but 8.9% lower for IN seed (P < 0.05). Similarly, final emergence in the field was higher for EN seed (EN: 48 ± 1.0%, CTRL: 35 ± 1.0%) with IN increasing emergence by 4% compared to the CTRL (P < 0.05).

As with *M. stipoides*, germination of *R. geniculatum* was higher for EN seeds (68 ± 1.5%) with the lowest for IN seeds (51 ± 1.4%) (CTRL: 58 ± 1.5%). However, there was no difference in seedling emergence in response to seed treatment under field conditions.

### Effects of salicylic acid on germination and field emergence

To assess the effect of SA, seeds that were provided SA (IS+ES) were compared to seeds that received the treatments without SA (IN+EN). If a significant difference was detected, SA delivery methods of encrusting (ES) and imbibing (IS) were then compared.

In *A. scabra* final germination of SA treated seeds (IS + ES), at 0.0 MPa water potential, was 4.3% lower than seeds treated without SA (IN + EN) (P < 0.05) (Fig 4A). At limited water availability of -0.6, -0.9, and -1.2 MPa SA treatments generally showed a slight but non-significant improvement in final germination. When tested in the field, SA treatments did not affect

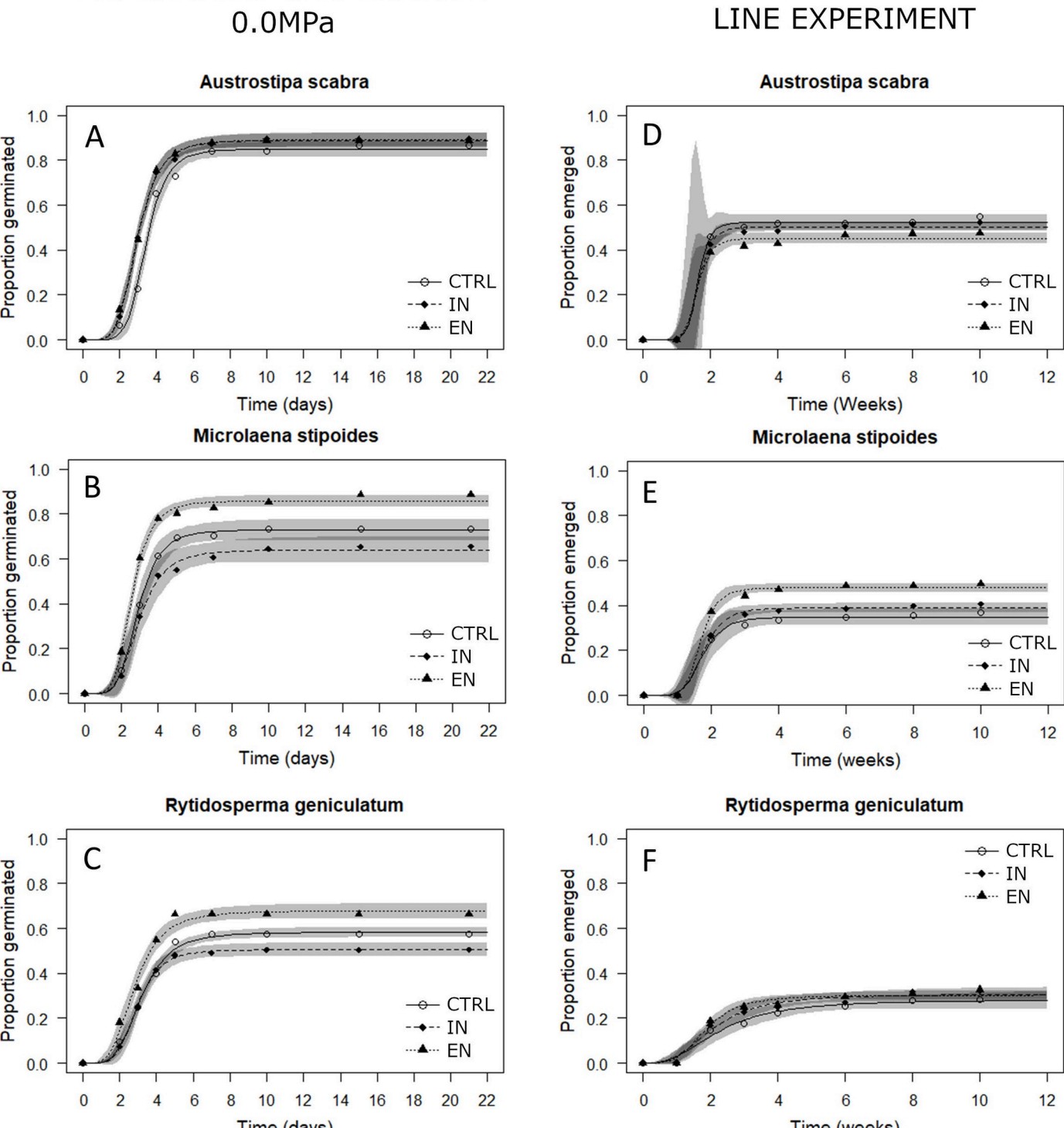

**Fig 3. Seed treatment germination and emergence curves.** Cumulative germination/emergence percentage curves for the three different seed treatments: Untreated (CTRL), encrusted (EN), and imbibed (IN) across the three species tested (*Austrostipa scabra*, *Microlaena stipoides*, *Rytidosperma geniculatum*). The lines represent cumulative germination over time. Data points are the germination recorded on a specific day/week and the shaded areas represent the 95% confidence intervals. A, B and C germination experiments were in controlled laboratory condition. D, E and F are seedling emergence in the field trials.

**Fig 4. Salicylic acid final germination and emergence.** Final germination and emergence of untreated seeds (CTRL), seed treated without salicylic acid (IN+EN) and seed treated with salicylic acid (IS+ES). A, B and C shows the laboratory germination experiment in petri dishes at 20°C at different water potentials (X axis). D and E show the germination and emergence results in field experiment 3 and 10 weeks after

sowing respectively. The species are listed in the X axis (Aus = *Austrostipa scabra*, Mic = *Microlaena stipoides*, Ryt = *Rytidosperma geniculatum*). Results followed by the same letter for the Water potential (lab experiment) and species (Field experiment) are not statistically different at p < 0.05.

germination (Fig 4D) but reduced final emergence (IN+EN: 51 ± 1.1%, IS+ES: 44 ± 1.1%, P < 0.001) (Fig 4E). SA encrusted seed (ES) emerged 5.6% lower than SA imbibed seeds (IS).

Similarly, *M. stipoides* germination at full water availability was reduced in the SA treated seed by 7.9% (P<0.05) (Fig 4B). SA delivered through encrusting resulted in higher germination (77 ± 2.1%) than SA delivered through imbibition (57 ± 2.2%). Under the limiting water potential of -0.6 MPa, germination for SA treated seed was improved from 77% ± 1.9% (CTRL) to 86 ± 1.9% (IS+ES), and encrusting (ES) allowed for a 12.7% increase in germination compared to imbibing (IS). However, at lower water potentials, SA treatment (IS+ES) reduced final germination by 5.6% (P < 0.05) at -0.9 MPa and by 11.2% (P < 0.01) at -1.2Mpa compared to the respective CTRL. In both situations encrusting allowed for higher germination then imbibition. Field germination and emergence of *M. stipoides* were not significantly affected by SA treatment, but all treatments (IN, IS, EN and ES) had higher emergence than the untreated control (Fig 4E).

When final germination was tested on *R. geniculatum*, no significant difference between seed treated with and without SA was detected at full and reduced water availability (Fig 4C). The only effect of SA was a delay in germination at 0.0MPa of 0.4 days. Field germination was no different for seed treated with and without SA; however, both treatments had lower germination than the untreated control. There was no difference in field emergence between seeds treated with and without SA. However, seed treated without SA (IN+EN) had significantly lower germination (P < 0.05) than CTRL. The raw data of the laboratory germination experiment, with information on seed mortality and germination without adjustment for viability, are provided in the S3 Appendix (Germination experiment results) in S3 File. The results of the analysis of the laboratory germination and field emergence in the line experiments are provided in the S4 Appendix (Germination Emergence Analysis) in S4 File.

### Effect of salicylic acid on survival and plant growth in field site conditions

In the line experiment the survival of plants that emerged from untreated seed (CTRL) was 32.3% for *A. scabra*, 41.2% for *M. stipoides* and 42.6% for *R. geniculatum* (Fig 5A). Plants emerging from SA treated seed (IS+ES), compared to seeds treated without SA (IN+EN), had a significantly (P > 0.001) increased survival by 12.9% in *A. scabra*, 13.5% in *M. stipoides* and 11.8% in *R. geniculatum*. In *A. scabra*, SA delivered through encrusting (ES) improve survival by 9.8% (P > 0.001) compared to SA delivered through imbibing (IN). In *M. stipoides* and *R. geniculatum*, no difference was detected between SA delivery systems on plant survival.

In the plot experiment, the average survival of seedlings in CTRL was of 82.5% for *A. scabra*, 82.5% for *M. stipoides* and 77.5% for *R. geniculatum* (Fig 5B). Survival was significantly improved (P < 0.01) by 8.2% and 15% in SA treated (IS+ES) *M. stipoides* and *R. geniculatum*, respectively, compared to SA untreated seeds (IN+EN). respectively and in *A. scabra*, survival was improved by 6.25% as a result of SA treatment but the difference was not significant. SA delivered through encrusting (ES) provided slightly better but non-significant survival. For both *M. stipoides* and *R. geniculatum*, SA treatment (IS+ES) improve survival by 17.5% and 10% respectively, compared to seed treated without SA (IN+EN) (Fig 5).

To evaluate treatment effects on plant growth plant height and above ground dry biomass were recorded. In *A. scabra*, no significant difference was detected between SA (IS+ES) and non-SA treatments (IN+EN) in either measurement. For *M. stipoides*, plant height for SA

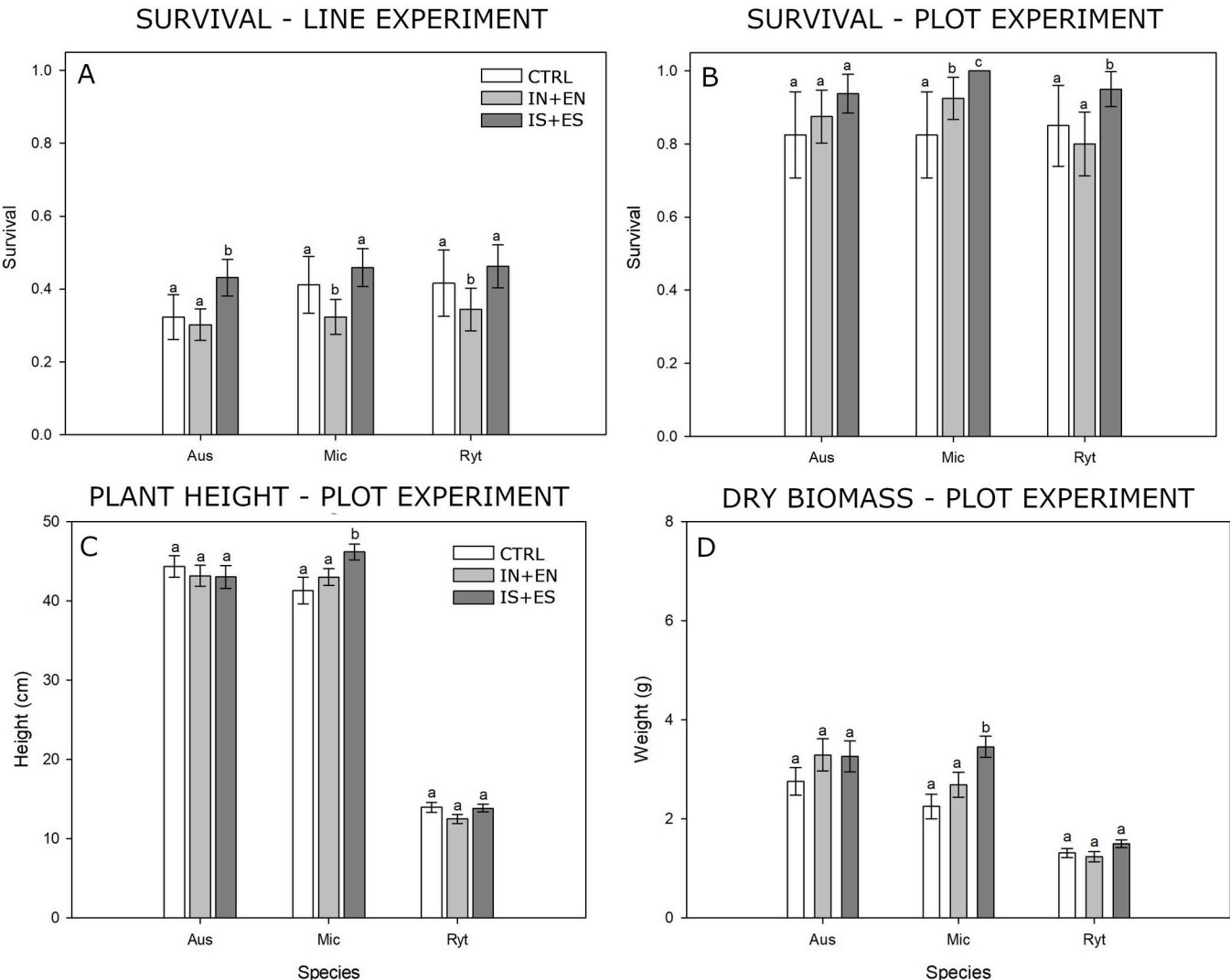

**Fig 5. Survival and plant growth.** Survival and plant growth comparison 40 weeks after sowing, between untreated seeds (CTRL), seed treated without salicylic acid (IN+EN) and seed treated with salicylic acid (IS+ES). Average plant survival in (A) line experiment with seeds sown alongside a 1 m line and in (B) plot experiment, where intraspecific competition was limited, by removing excess seedlings and leaving 10, evenly spaced seedlings per 0.25 m $^2$ plot. (C) Average height and (D) biomass of plant collected from the plot experiment. Results followed by the same letter are not statistically different at p < 0.05.

treated seed was significantly improved (P < 0.05) from 41 cm ± 1.7 cm (CTRL) and 43 cm ± 1.0cm (treated seed without SA IN+EN), to 46 cm ± 1.0 cm. Dry above-ground biomass was also higher in SA treatment (IS+ES, 3.4 g ± 0.22g) compared to non-SA treatments (IN+EN, 2.7 g ± 0.25 g) and untreated controls (CTRL, 2.2 g ± 0.25 g) (both P < 0.05). In *R. geniculatum*, there was no significant difference in height between treatments. Dry biomass for SA treatment (IS+ES, 1.5 g ± 0.08g) was significantly higher (P > 0.05) than treated without SA (IN+EN, 1.2 g ± 0.10g), but not significant compared to the untreated control (CTRL, 1.3 g ± 0.09 g). In terms of plant growth, no significant difference was detected in the study species between SA delivery through imbibing (IS) or encrusting (ES). Raw data for field bag germination, line and plot experiment are provided in S5 Appendix (Field experiments results) in S5 File.

## Discussion

Testing the first hypothesis concerning the effects of seed treatment process (CTRL vs IN vs EN) on germination and emergence, *A. scabra* showed no treatment effect as was hypothesised. Instead, *M. stipoides* and *R. geniculatum* showed unexpected significant differences between treated seeds (imbibed and encrusted) and the control. In the germination experiment, the two species behaved similarly, with encrusted seeds performing better than controls, while imbibition had negative effects on both final germination and germination speed.

Contrary to what was initially postulated in the second hypotheses, SA application did not clearly improve seed germination and emergence in the field or in controlled laboratory condition across a water availability gradient on the tested species. Even though significant differences across treatments in laboratory germination were at times detected (for example in *M. stipoides* at all water potentials), there was no clear trend suggesting that SA application could consistently increase germination. Similarly, in the field germination bag experiment no significant improvement was detected for SA treated seeds across all species. In the field emergence experiment, SA application did not affect outcomes for *M. stipoides* and *R. geniculatum* but a significant drop was observed for *A. scabra*. This might suggest that in *A. scabra*, the interaction of SA treatment with unidentified variables present in the soil at field site might have triggered a negative response. When a difference in germination and emergence was detected for seed treated with SA, encrusted seed performed slightly better than imbibed seeds. However, this difference is most likely due to the process itself, as highlighted previously, rather than the efficacy in delivering SA.

The third hypothesis of SA improving field survival and growth was partially confirmed by the results of the field line and plot experiments. Plant grown from seeds treated with SA resulted in increased height and biomass production in *M. stipoides* whilst no differences were observed in the other two species (*A. scabra*, *R. geniculatum*). Salicylic acid provided a significant improvement in plant survival in both line and plot experiments. Although response among species varied, with the least effects detected in *A. scabra*, the overall trend showed marked benefits in term of survival from SA-treated seeds. The improved survival at this stage could be explained by the already described stress resistance proprieties of SA [44]. This improvement in survival might be due to a variety of factors, such as the effect of SA in mediating reactive oxygen species (ROS) and triggering defence-related processes [51], and its effect on productivity and growth [52]. In this study, just one of the three species tested (*M. stipoides*) showed a higher biomass production as a response to SA treatment. A previously published study reported that externally applied SA had increased root development [53], but root growth was not evaluated in this study. Nevertheless, as this study shows, the effects of exogenous SA delivery are still present months after its application. Salicylic acid absorbed through the seed (imbibing), or through emerging radicle and roots (encrusting) could be converted in SA glucoside and transferred in the vacuole for storage [54]. SA glucoside could be mobilized and moved through the plant after been converted in methyl salicylate, and eventually turned back to SA when needed [24].

When SA delivery mechanisms of imbibing and encrusting were compared in terms of improving plant survival, a significant difference was rarely detected, suggesting that seed encrusting could be used to deliver SA and its stress resistance inducing proprieties. The advantage of using SA via seed coating over imbibition lies in the capability of storing seed after treatment. Seed imbibition can trigger a seed priming effect that could improve germination speed and synchronicity in the short term [55], but such imbibition could accelerate seed ageing processes, reducing seed shelf-life and storability [56]. Another advantage of seed coating over imbibition is that while it delivers SA stress resistance, it can also improve seed

handling and sowability, along with delivering a wide variety of active ingredients, such as protectants, micronutrients, germination promoters, and microorganisms [21].

## Effect of intraspecific competition on plant survival

A potentially significant, yet unintended, result of this experiment is the difference in plant survival between the line (high seedling density) and plot (low seedling density) experiment (Fig 5). Regardless of treatment, survival in plot experiment was more than double that in the line experiment across all species. Even though these two experiments are not directly comparable, the differences in the two competition scenarios might provide useful insight relevant to restoration practitioners. According to the seedling emergence data, the seedling density in the line experiment was of 520 seedling/m$^2$ in *A. scabra*, 430 seedling/m$^2$ in *M. stipoides and* 280 seedling/m$^2$ in *R. geniculatum*, whilst for the plot experiment seedling density was 40 seedling/m$^2$ across all species. Based on personal observations, the plants with limited competition (plot experiment) were generally more developed before summer than the ones at higher competition (line experiment), probably resulting in deeper and broader root systems which allowed access to water during the dry summer months, ultimately increasing chances of survival. These results suggest that intraspecific competition within these species could play a major role in seedling establishment rate. This factor needs to be taken in consideration when planning for seeding operation. However, further studies designed to directly compare the effect of seeding density on plant survival could provide restoration practitioners with more accurate seeding rate prescriptions that could help limit wastage of valuable and expensive seeds [57].

## Demographic processes

A study evaluating the demographic processes that influence established of native species from seed in field trials, performed by James et al. (2011) [12], identified the main bottleneck to successful seedling recruitment as the emergence phase (when germinated seeds failed to express surface emergence). However, in our experiment the drop between germination and emergence was relatively small with the probability of emergence from germinated seed ranging from 0.92 in *A. scabra* to 0.61 in *R. geniculatum* (Fig 6). This trend might be due to the favourable climatic and soil conditions during the year the study was conducted, with average night and daily temperature ranging between 10˚C and 18˚C, and maintained soil moisture content of 0.08–0.18m$^3$/m$^3$ (water potential range between -0.2 and -0.7 MPa) during the first month after sowing, when most of the emergence occurred. These conditions have not allowed for the detection of the stress reduction proprieties of SA that were originally hypothesised at the germination and emergence phase. However, the field data, combined with the controlled germination experiment with reduced water availability, suggest that SA might not affect seed performances at the establishment phase, as suggested by Xie *et al*. 2007 [31]. Further studies are needed to test this hypothesis under more severe stress conditions and on different species.

Significant effects of SA delivering stress resistance were instead detected on the survival of established plants over the summer when seedlings had to endure prolonged periods with little access to water. Total precipitation between November 2017 and February 2018, removing two major rainy events that happened over a few hours (60 mm on December 20[th] and 147 mm on January 18[th]) were less than 30 mm (Fig 2). The effects of the dry summer months were evident in the field line experiment, with the probability of plant survival from an emerged seedling being 0.32 for *A. scabra*, 0.41 for *M. stipoides* and 0.42 for *R. geniculatum*. In this case, SA treated seed survived significantly better than the seed treated without SA for the three species. When considering the cumulative survival from the number of seeds initially sown, SA treatment provides a significantly higher number of successful plant establishment

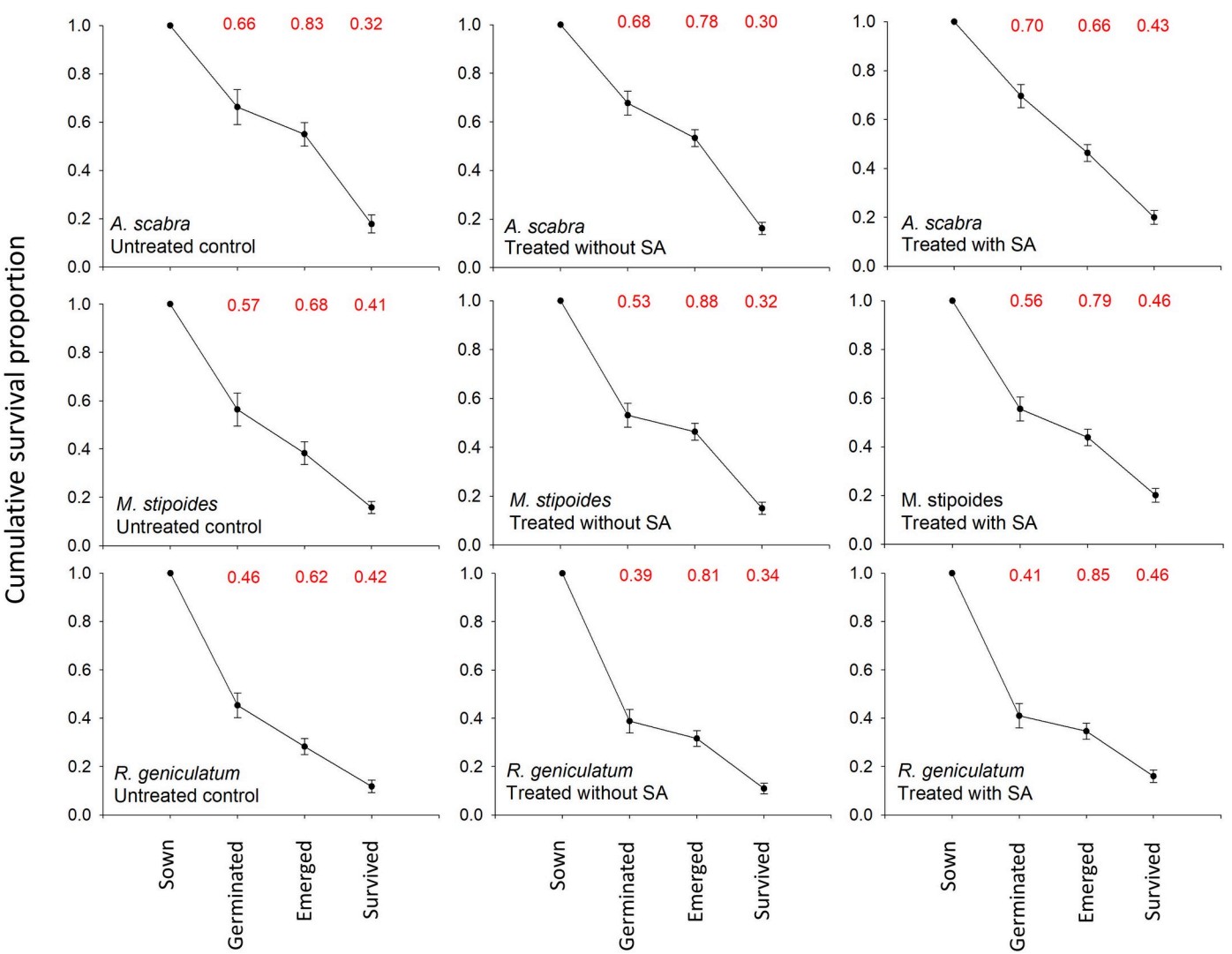

**Fig 6. Cumulative survival proportion.** Demographic process through various life stages for the three species tested without treatment, treated without SA (IN + EN) and treated with SA (IS + ES). On the top of each graph, in red, are reported the probability of transitioning between life stages. This demographic data are based on the "in line" experiment whereas seedlings were not removed after emergence.

events, even for *A. scabra*, in which emergence of SA treated seed was lower than the seed treated without SA. The improved establishment of Australian grasses treated with SA suggest that a similar approach can be tested in different systems. For example, in cold desert restoration, where high seedling mortality occurs in the winter months due to a variety of biotic and abiotic stressors such as freezing temperature, drought and pathogens, [12,58] SA application to seed prior to seeding could be a useful adjuvant to promote seedling survival.

## Conclusions

This study showed that seed coating with salicylic acid can improve survival of three Australian native grass species used in restoration programs. However, further research is needed to evaluate SA coating capability to improve native plant resistance to different biotic and abiotic stresses such as extreme temperatures, salinity, pathogens, and herbicides. Moreover, coating with SA in combination with other beneficial compounds should be tested on restoration

priority species, because their combined impact on seed germination, emergence, growth and plant establishment could improve the successful deployment of native seed onto degraded landscapes, ultimately allowing for a more efficient seed-based restoration.

## Supporting information

**S1 File. Layout of the experimental design for the field experiments of "germaition bags", "in line" and "in plots".**
(PDF)

**S2 File. Soil moisture retention curve measured with a Dew Point potentiometer WP4C on soil samples collected from the site where field experiments were conducted, and average weekly volumetric soil water content recorded with a HOBO Micro Station Data Loggers (Onset Computer Corporation, Bourne, MA, USA) at the field site during the first 12 weeks of the experiments.**
(PDF)

**S3 File. Raw data of the laboratory germination experiment.** Summary tables are provided for each species with the averages and standard errors for mortality, germination, and germination adjusted for viability.
(PDF)

**S4 File. Final germination and T50 values of laboratory germination experiments and emergence in the field line experiment.** Statistics obtained with parameter comparison of DRM model comparing treatment, SA, and combination of treatment and SA against the untreated control.
(PDF)

**S5 File. Raw data of the field germination experiments.** Summary table are provided for each experiment with average and standard errors for germination (germination bag experiment), emergence (line experiment), and survival (line and plot experiments).
(PDF)

## Acknowledgments

We acknowledge that the research undertaken and presented here was done *on the lands of the Gnaala Karla Booja peoples of the Noongar Nation* and pay our respects to their elders, past, present, and emerging.

This study is dedicated to the memory of the late Dr Tissa Senaratna who was instrumental in establishing salicylic acid as a key principle for improving plant growth and development and without whom this and many other studies would not have been possible.

We would like to acknowledge Dr. Adam Cross for the edits and comments provided to the manuscript and Native Seed Pty Ltd for donating the seeds used in this study. We would like to thank the reviewers whose thorough comments and recommendations greatly improved quality and clarity of the manuscript. A special thanks goes to Sadichhya Adhikari, for the support provided in editing and imporving the reviewed version of the manuscript.

## Author Contributions

**Conceptualization:** Simone Pedrini, Jason C. Stevens, Kingsley W. Dixon.

**Data curation:** Simone Pedrini.

**Formal analysis:** Simone Pedrini.

**Funding acquisition:** Kingsley W. Dixon.

**Investigation:** Jason C. Stevens, Kingsley W. Dixon.

**Methodology:** Simone Pedrini, Jason C. Stevens, Kingsley W. Dixon.

**Project administration:** Kingsley W. Dixon.

**Software:** Simone Pedrini.

**Supervision:** Jason C. Stevens, Kingsley W. Dixon.

**Validation:** Kingsley W. Dixon.

**Visualization:** Simone Pedrini.

**Writing – original draft:** Simone Pedrini.

**Writing – review & editing:** Simone Pedrini, Jason C. Stevens, Kingsley W. Dixon.

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
