## [Decision Letter · Decision Letter 0]

18 Dec 2020

PONE-D-20-33303

Seed encrusting with salicylic acid: a novel approach to improve establishment of grass species in ecological restoration

PLOS ONE

Dear Dr. Pedrini,

Thank you for submitting your manuscript to PLOS ONE. After careful consideration, we feel that it has merit but does not fully meet PLOS ONE’s publication criteria as it currently stands. Therefore, we invite you to submit a revised version of the manuscript that addresses the points raised during the review process.

As you can see, the conclusions reached by two independent reviewers are quite different. As you revise your manuscript and prepare it for submission, please consider the comments of both reviewers very carefully. It would be helpful if you could provide responses to the reasons given for rejecting the manuscript.

We look forward to receiving your revised manuscript.

Kind regards,

Craig Eliot Coleman, PhD

Academic Editor

PLOS ONE

Journal Requirements:

"S.P. was the recipient of a Curtin University International Postgraduate Research Scholarship. A.T.C is

 the recipient of the Research Fellowship in Restoration Ecology jointly funded by the EcoHealth

Network, Gelganyem Limited, and Curtin University. Seed were donated by Native Seed Pty Ltd. This

research was supported by the Australian Government through the Australian Research Council

Industrial Transformation Training Centre for Mine Site Restoration (Project Number ICI150100041).

The views expressed herein are those of the authors and are not necessarily those of the Australian

Government or Australian Research Council."

"SP and AC were supported by the Australian Research Council Industrial Transformation

Training Centre for Mine Site Restoration (Project Number ICI150100041). The funder did not play any role in the study design, data collection and analysis, decision to publish, or preparation of the manuscript?"

Additionally, because some of your funding information pertains to commercial funding, we ask you to provide an updated Competing Interests statement, declaring all sources of commercial funding.

In your Competing Interests statement, please confirm that your commercial funding does not alter your adherence to PLOS ONE Editorial policies and criteria by including the following statement: "This does not alter our adherence to PLOS ONE policies on sharing data and materials.” as detailed online in our guide for authors  http://journals.plos.org/plosone/s/competing-interests.  If this statement is not true and your adherence to PLOS policies on sharing data and materials is altered, please explain how.

Please include the updated Competing Interests Statement and Funding Statement in your cover letter. We will change the online submission form on your behalf.

3. Please ensure that you refer to Figures 4 and 5 in your text as, if accepted, production will need this reference to link the reader to the figure.

Reviewers' comments:

Reviewer's Responses to Questions

**Comments to the Author**

1. Is the manuscript technically sound, and do the data support the conclusions?

Reviewer #1: No

Reviewer #2: Partly

2. Has the statistical analysis been performed appropriately and rigorously? 

Reviewer #1: I Don't Know

Reviewer #2: Yes

3. Have the authors made all data underlying the findings in their manuscript fully available?

Reviewer #1: No

Reviewer #2: Yes

4. Is the manuscript presented in an intelligible fashion and written in standard English?

Reviewer #1: No

Reviewer #2: Yes

5. Review Comments to the Author

Reviewer #1: Pedrini et al carried out a study entitled with salicylic acid: a novel approach to improve establishment of grass species in ecological restoration”.The research paper focuses on the application of salicylic acid (SA) to native grass seeds to improve a range of life stage transitions including germination, emergence and survival. The need for improved native seed regeneration is paramount and experimental work such as this is critical to move this industry forward in a positive manner. The authors attempted to utilise some well-known SA treatments that have been used in agriculture into the native seed industry focussed on restoration. They chose two main approaches of applying SA to seeds via soaking/imbibing or through addition in artificial seeds coats. No major findings were highlighted under lab germination assessments or for emergence phases. The main conclusions of this work were the suggested benefit of SA treated seeds in survival and growth under field conditions.

However, I have some concerns of the experimental design that would call the authors conclusions into question. First, the application of SA via imbibing or encrusting was un-balanced and not all treatments were tested in each individual experiment presented in the results (e.g. only a sub-set of the five treatments listed below were tested in Figure 3 and then a separate set in Figure 4/5). With this, some conclusions of the benefits of SA cannot be concluded as other factors could have been influencing these results (e.g. the coat alone). Second, there is an issue of pseudo-replication and installation of some of the germination bag, line experiment, and plot experiment in the field trial. Some of this set-up raises concerns for how some of the findings are discussed. Further commentary for these two components are outlined below in the Line by Line feedback.

At times, the grammatical standard of this paper lacked attention to detail, and many in-text links to Figures were missing making it very difficult to carry out a detailed review of the data and main findings. The naming convention for each treatment was also unclear and I was constantly pausing to understand which treatment was which and therefore could not examine the finer details of the study design and findings with clarity. It would help at first introduction to name all these treatments and keep them consistent throughout the paper (notes made in the Line by Line comments).

Therefore, for this paper to be suitable for publication a major re-write is to all sections of the manuscript. Some further comments/suggestions are located below that were noted during my revision.

(*note* apologies for the length of some of this review – I have attempted to provide as much feedback as possible in the hope that it improves the manuscript. Please don’t be offended if the tone comes across as short/abrupt – this is an artefact of very little time and a busy schedule).

L 7 - Capitals missing from school name

L 14 - What is the difference between affiliation one and five here?

L 29 - Seed coating has been developed for many years. Unsure why the authors are saying "first-time" here.

L 35 – “conditions”

L 41 - These results presented here are very unclear. (1) the sentence suggests you are discussing seed sowing numbers at low and high densities. Therefore, the numbers reported here should be reported in the number of seeds sown not plant per m2. Or (2) your results are referring to the established plants numbers. If so this does not make sense and contradicts your findings.

L 59 - Delete first "like"

L 59 - “landscapes”

L 60 – ref 9 - Is this a suitable reference. Surely there are specific mining examples.

L 71 – after the reference bracket the end of the sentence makes no sense.

L 94 – in relation to the word ‘never’, this is not quite true. Work has occurred in native species before for restorative work (even involving some of the authors listed here). A simple search online brings up some work in Western Australia close to this paper's study area.

https://www.agrifutures.com.au/wp-content/uploads/publications/10-061.pdf

Stevens, J.C., Barrett-Lennard, E.G. and Dixon, K.W. (2006). Enhancing the

germination of three fodder shrubs (Atriplex amnicola, A. nummularia, A. undulata;

Chenopodiaceae): implications for the optimisation of field establishment. Australian

Journal of Agricultural Research 57: 1279-1289.

https://www.mla.com.au/download/finalreports?itemId=436

L 135 – the concentration chosen - is this sufficient for native seeds, for which you have said has ‘never’ been tested. One would think you would carry out a dose response curve experiment to optimise SA treatments of your native seed first??

L 140 – imbibition duration - Similar to the apparent lack of a SA dose-response assessment, how do you know this length of soaking time was suitable/optimum?

L 145-146 – “dusted onto the rotating mass” - on Lines 143-144 you stated that the filler material was dusted on using a paint brush. But here you state that the powder was dusted on while seeds were rotating. This appears near impossible as you would disrupt the seed flow in the coater. Please clarify

L 148 – “multiple seeds” - unclear. Do you mean seeds clustered together? Not singular?

L 155 – “PEG solution was added weekly” - at each re-hydration point was the water potential checked to ensure your target water potential was maintained. This is critical because at some higher concentrations of PEG the water potential changes as water evaporates, and then at re-hydration with your target concentration this then leads to an altered water potential.

L 158 - No mention of 0MPa for ‘control’ conditions.

L 165 – “Non-viable seeds were excluded from the total” – Does this mean all data presented in the lab trials are cut-test adjusted? If you did do this, you need to clearly state and show these results for readers to interpret. For example, what if your SA imbibing treatments (or encrusting) lead to a significant drop in viability but you only present the viable component in graphs. This could mislead the reading to more positive findings.

L 167 – site selection - Is it a grassland requiring restoration? The whole Intro was pitched around grassland restoration. It would aid the reader here to provide some more background to the field trial system.

L 169 - These mining operations from aerial image searches look like forested systems? If your focus is on grasslands please elaborate this mining example.

L 173 – “the five treatments” - Which five. Re-iterate.

L 175 - It is unclear here whether your "replicates" here are the same as blocks in the following Line. If they are, the field layout appears to be pseudo-replicated where you have only tested your treatment in one large block containing 4 replicates. It is not replicated at the site level. How have you accounted for this in your analysis. Only testing in one site would potentially confound your findings.

L 180 and 189 – related to the above comment it appears that your germination bags and plot experiment where sown outside of the replicated line experiment design leading to three independent experimental sub-sets (germ bags , line experiment and plot experiment). This confounds your experimental finding and conclusions making it difficult to directly compare (see comments later surrounding your conclusion of the density/competition work). This is one flaw of this work that needs addressing.

L 194 - Write 45 in words. You typically dont start sentences with numbers.

L 208 – “emergence” - This is confusing. All this is worded for germination then you have mentioned emergence here?

L 209 - Related to the above comment how does this work for emergence. This section needs revision to accommodate both life stages you are addressing. To the reader as it stands it is not clear what you have done.

L 215-216 - This is very unclear to the reader. I have drawn multiple diagrams noting each of your treatment and I still cannot work out what you mean here. Un-treated control = easy to work out. "Treated without SA" is this the "imbibed without SA" or the "coated without SA"? Same goes for the "treated with SA". I would request a complete re-write so that we can critically review your results/figures/tables.

L 232 – 243 - Grammar is very poor. Suggest a re-work

L 236 – 237 Is both Encr and Imb here without salicylic acid (SA). For now, this is not very clear and finding it hard to work out what treatment you are comparing to what. And discussed later on, your treatment naming convention needs further thought. It is very hard to keep track of what treatment is what. Here you have Encr and Imb, later on you have SA and No.

L 239 - P value was lower case in stats section. Be consistent.

L 242 - More frequent measures of emergence may have provided some less variable results here. There is a very high range of variation between the 1 week and 2 week monitoring points which is a shame as this could have been powerful.

L 246 - It is unclear here which significance level this is related to? Assume it is for the Encr v the Ctrl? Not this latter, minor 4% difference?

L 248 - Check wording here between the brackets. Some info for Ctrl is missing.

L 252 – 254 - This is best suited to the methods section. This also relates to my query on Line 215 – 216 about knowing what you are comparing. Unclear treatment names also add to this difficulty.

L 255 - This is a discussion point

L 252 – 256 - Now that I have read on a little more into your results here it is very unclear what treatment you are comparing to what.

As I read it you are comparing untreated (Ctrl) seeds to (1) seeds that received SA via imbibition, but not encrusted ("NO") and (2) seeds that DID NOT receive SA via imbibition, but were received SA via encrusting ("SA" in your figures")

If this is true, then this small sub-trial work is unbalanced and slightly confounded. You are missing a treatment of untreated seeds that are encrusted without SA imbibition and seeds that were imbibed with SA and encrusted. One example of the confounding factor here is that one could argue that just encrusting seeds alone could provide improved water relations for seed germination at lower water potentials. As you do not have this treatment comparison to compare all findings are speculative.

Line 257 - What is this statistical result referring to. If it is for Cntrl v SA then according to your post-hoc lettering above the germination columns at 0 MPa this is not true. They both share a small lower case "b". Recommend checking your numbers. Imbibed in water, if this is what 'NO' is (very hard to tell!), then there appears to be a significant increase in water imbibition when compared to SA imbition

L 258 - Quality control on what has been submitted? Assume this is for figure 4?

L 261 – wording in brackets - What is this being compared to? There is not mention here of the Ctrl.

L 262 - Many in-text figure citations are missing making it hard to link the results to the figures.

L 263 – I suggest to not use the word optimal for your MPa conditions. This is an assumption that non water limited conditions is optimal.

L 264 – ‘SA delivered through encrusting…’ - Example of confounded data here. What would happen if you just encrusted the untreated seed? Data in Figure 3 (your previous work suggest the M stipoides benefits from encrusting alone - ca. 80%+ germination). But you do not have this comparison here under your water potential experiment here to conclude how it would perform.

L 285 - Which Figure am I looking at now? Assuming 5C but I should have to work it out.

L 286 – compared to seeds treated without SA - so are you referring to your "NO" treatments. This is very hard to follow. Recommend a major updated to your treatment naming convention. If your methods you just word that you have five treatments. Then at times we have "Ctrl", "No", "SA" and even "ES" and "IS" on Line 254 than I still haven't seen.

L 335-337 – “A potential explanation….(petri dish).” - This is highly speculative and an odd discussion point. First, soaking seeds for 24 hours in either water or water containing beneficial agents is a well-established technique for use in native seeds. Even some of the authorship group here have published many papers in this space. So, I find the suggestion of anoxic stress during 24h of soaking to be off track. Further there is no measured evidence provided in this study to suggest this could be present. Second, the suggestion of anoxic environments due to extended time spent in a petri dish is even more out there. International seed testing standards/protocols for germination testing in both agriculture and native seeds use Petri dish work to carry out these standard assessments and have been standard practice for centuries. For instance, the classic Baskin and Baskin book reports germination data for 15,000+ species largely from Petri dish work. ISTA seed testing guidelines recommend the same approach. Even the authorship group have just set some standards in this very space (Pedrini and Dixon 2020, Rest Ecol, "Guidance Statement 13.2:). I find this a nonsensical discussion point and it should be removed or heavily backed up by evidence using native seeds (most evidence reported below this point is agricultural).

L 338 – oxygen availability - Same as previous comment. All speculation.

L 341 – 342 - Same as previous comment. All speculation.

L 352-353 - One key step that has been missed in this study is some detailed preliminary work on the concentrations of SA required and duration of soaking needed to test whether there will be a benefit at all for this use in native seeds. The authors clearly stated "However, it has never been tested on native species for ecological restoration", Line 94-95, yet they jumped into testing one concentration and one soaking time that worked on agricultural seeds. Then most of this discussion hovers around maybe the concentration could be altered. This study would have been strengthened by detailed dose-response research into the most suitable SA treatment and soaking duration.

L 366-368 – “SA also provided….without interspecific competition” - As discussed previously, this finding would also be confounded by the experimental design of this study. It severely limits the interpretation provided here. For instance, the discussion point here is saying that SA provided a significant improvement in plant survival in both competition scenarios, yet, without a fully balanced treatment design the authors cannot outright conclude that SA provided all this benefit. What would happen is just encrusted (no SA) were tested? Without it these results and conclusions are flawed.

L 376-379 - Once again, speculative and not backed up by data. From an experimental design perspective, the low and high density are not directly comparable. They are independent assessments that were set-up very differently. The 'line experiment' was done in 1m long strips by 5cm wide, sowing 100 seeds (roughly equating to one seed per 5cm2) - presumably in the replicated plot design outlined on lines 173-176. Whereas, the 'plot experiment' was done in 50 x 50cm2 plots, sowing of 100 seeds over this 2500cm2 (1 seed per 25cm2) seeded in plots outside of the line experiment area. So straight up, the plot design (line v plot) and thinning protocols are different (e.g. the plot experiment was thinned at 4 weeks, but nothing was done at the line experiment). Further, as the seeded areas were in different locations, site level variation could have driven these differences observed. Replicate numbers for the 'plot' study are not mentioned anywhere that is obvious. To be truly comparable it would have been best to have this assessment carried out with the same sowing technique - either as line or plots - with known differences in seed sown at Day 0 or thinned at a point down to target densities. This would make a direct comparison possible. As it stands, again the design in problematic.

L 392- 393 - Was the Hobo logger calibrated for the soil type of the field site? Where is this data presented? Where any soil analyses completed?

L 399-400 - See discussion points mentioned previously aligned to the unbalanced treatment design and confounding concerns. Not sure if this SA significant effect can be set in stone.

L 403-404 - How can you state that there was a summer drought when the site got close to 210mm over the summer period!

L 455 – many reference need checking for complete information or format. E.g. ref 14 is LOUD. How could I find ref 38? Check out ref 49

Figure 1 – Austrostipa and Rytidosperma both not spelled correctly.

Figure 4 – check spelling again for Austro

Figure 6 – A. stipa?????????

Supp Material file - species spelling again

Reviewer #2: Pedrini et al. research provides a valuable contribution to the field of restoration ecology by demonstrating that salicylic acid can be applied within a seed coating to improve drought tolerance of important grass species in Australia. This work also has implications for improving seeding efforts in dryland systems across the globe.

The authors need to improve the clarity in their methods section. For example, I would like to see a better description on why the different studies were performed and how they related to their hypotheses. Increased detail and accuracy could also be given in describing the seed treatments used in the different studies and how the results were analyzed. They state that five treatments were tested in all studies but the field study results only have three seed treatments.

I had a number of suggestions throughout the manuscript to improve grammar, clarity, and presentation of the study (see attached pdf). My edits were not exhaustive but show examples of issues that should be considered. The authors also need to write a conclusion. After some work this manuscript will be a quality publication.

6. PLOS authors have the option to publish the peer review history of their article (what does this mean?). If published, this will include your full peer review and any attached files.

Reviewer #1: No

Reviewer #2: **Yes: **Matthew D. Madsen

---

## [Author Response · Author response to Decision Letter 0]

1 Feb 2021

Dear editor and reviewers

Thank you for the numerous and detailed feedback on the manuscript. I have to acknowledge that in its original form the manuscript lacked clarity and consistency in presenting the methods and results. I have tried to follow the reviewer’s advices and hopefully, in its current form, the manuscript is easier to read, understand, and review. Large parts of the paper have been drastically modified, and the structure has been changed. I have tried to respond to all of the comments, either by modifying the paper or providing explanations as to why the suggested changes were not made. 

In the following pages, I have provided the rebuttal to each comment. A file with the same responses (but in red font to differentiate the rebuttal from the reviewer's comments) has been submitted. I hope you will find the revised version of the manuscript easier to understand and interpret. 

Kind regards

Simone Pedrini

Reviewer #1: Pedrini et al carried out a study entitled with salicylic acid: a novel approach to improve establishment of grass species in ecological restoration”. The research paper focuses on the application of salicylic acid (SA) to native grass seeds to improve a range of life stage transitions including germination, emergence and survival. The need for improved native seed regeneration is paramount and experimental work such as this is critical to move this industry forward in a positive manner. The authors attempted to utilise some well-known SA treatments that have been used in agriculture into the native seed industry focussed on restoration. They chose two main approaches of applying SA to seeds via soaking/imbibing or through addition in artificial seeds coats. No major findings were highlighted under lab germination assessments or for emergence phases. The main conclusions of this work were the suggested benefit of SA treated seeds in survival and growth under field conditions.

However, I have some concerns of the experimental design that would call the authors conclusions into question. 

First, the application of SA via imbibing or encrusting was un-balanced and not all treatments were tested in each individual experiment presented in the results (e.g. only a sub-set of the five treatments listed below were tested in Figure 3 and then a separate set in Figure 4/5). With this, some conclusions of the benefits of SA cannot be concluded as other factors could have been influencing these results (e.g. the coat alone). This issue was raised by both reviewers. All treatments were tested in each experiment, but in presenting the data I decided to group different treatment with the intentions to improve clarity and address each hypothesis. However, this approach was not clearly explained in the methods and results section. These two sections have been rewritten and the treatment naming standards and treatment grouping updated throughout the document. Raw data that presents all the treatment tested have also been provided in appendix 3 and 5.

Second, there is an issue of pseudo-replication and installation of some of the germination bag, line experiment, and plot experiment in the field trial. Some of this set-up raises concerns for how some of the findings are discussed. The experimental design is now discussed in more detail in the methods section and I have provided supplemental material with the layout of the field experiments that should clarify the issue of pseudo-replication (appendix 2). When originally set up, the three field experiments were not meant to be directly compared and the results were analysed and presented separately. However, I found that some trends across these experiments, such as the effect of intraspecific competition and demographic processes, could be relevant to restoration practitioners and would be a good starting point for further testing of seeding rate in restoration scenarios. In the discussion, we have now separated this section from the discussion of the main results and acknowledge the limitation of the experimental design.

Further commentary for these two components are outlined below in the Line by Line feedback.

At times, the grammatical standard of this paper lacked attention to detail, and many in-text links to Figures were missing making it very difficult to carry out a detailed review of the data and main findings. Thorough grammatical check was performed on the documents and in text link to figures added.

The naming convention for each treatment was also unclear and I was constantly pausing to understand which treatment was which and therefore could not examine the finer details of the study design and findings with clarity. It would help at first introduction to name all these treatments and keep them consistent throughout the paper (notes made in the Line by Line comments). Name convention for the treatment has been rewritten and checked for consistency across the document

Therefore, for this paper to be suitable for publication a major re-write is to all sections of the manuscript. Some further comments/suggestions are located below that were noted during my revision.

(*note* apologies for the length of some of this review – I have attempted to provide as much feedback as possible in the hope that it improves the manuscript. Please don’t be offended if the tone comes across as short/abrupt – this is an artefact of very little time and a busy schedule).

L 7 - Capitals missing from school name Fixed

L 14 - What is the difference between affiliation one and five here? Five removed.

L 29 - Seed coating has been developed for many years. Unsure why the authors are saying "first-time" here. Removed.

L 35 – “conditions” Fixed.

L 41 - These results presented here are very unclear. (1) the sentence suggests you are discussing seed sowing numbers at low and high densities. Therefore, the numbers reported here should be reported in the number of seeds sown not plant per m2. Or (2) your results are referring to the established plants numbers. If so this does not make sense and contradicts your findings.

The sentence is referring to seedling density, not seeding rate. Seeding rate was removed and the sentence corrected as follow: “Seedling survival over the dry summer season was more than double at low seedling density (40 plants/m2) compared to high density (380 plants/m2).” Given that in the following sentence I specify “adjustment of seeding rate according to expected emergence” I don’t’ think there is a contradiction.

L 59 - Delete first "like", deleted

L 59 - “landscapes”, fixed

L 60 – ref 9 - Is this a suitable reference. Surely there are specific mining examples. Replaced with “Stevens J, Dixon K (2017) Is a science-policy nexus void leading to restoration failure in global mining? Environmental Science & Policy 72:52–54”

L 71 – after the reference bracket the end of the sentence makes no sense. Removed

L 94 – in relation to the word ‘never’, this is not quite true. Work has occurred in native species before for restorative work (even involving some of the authors listed here). A simple search online brings up some work in Western Australia close to this paper's study area. By “never” is meant the application of SA via seed coating to natives for restoration. To make this point clearer the sentence has been modified as follow. “However, the application of SA via seed coating has never been tested on native species for ecological restoration.” 

https://www.agrifutures.com.au/wp-content/uploads/publications/10-061.pdf

Stevens, J.C., Barrett-Lennard, E.G. and Dixon, K.W. (2006). Enhancing the

germination of three fodder shrubs (Atriplex amnicola, A. nummularia, A. undulata;

Chenopodiaceae): implications for the optimisation of field establishment. Australian

Journal of Agricultural Research 57: 1279-1289

https://www.mla.com.au/download/finalreports?itemId=436

In the saltbush paper, and other similar reports (see below), the main focus was the evaluation of native perennial for pasture more than restoration. In all the experiment SA was delivered in growing media or via imbibition. Following this suggestion, I have decided to mention and reference the papers/reports as follows.

When tested on Australian native seeds for pasture, in some species, germination under drought and hyper salinity conditions was enhanced by the application of SA [37–39]. 

37. Stevens JC, Barrett-Lennard EG, Dixon KW. Enhancing the germination of three fodder shrubs ( Atriplex amnicola , A. nummularia , A. undulata ; Chenopodiaceae): implications for the optimisation of field establishment. Aust J Agric Res. 2006;57: 1279–1289. doi:10.1071/AR06031

38. Clarke S, Stevens J, Ryan M, Mitchell M, Chivers I, Dixon K. Native Perennial Grasses for Sustainable Pasture Systems. KINGSTON ACT 2604; 2010. 

39. Nichols P. Reliable Establishment of Non-Traditional Perennial Pasture Species. North Sydney; 2011. Available: https://www.mla.com.au/download/finalreports?itemId=436

L 135 – the concentration chosen - is this sufficient for native seeds, for which you have said has ‘never’ been tested. One would think you would carry out a dose response curve experiment to optimise SA treatments of your native seed first?? Previous personal experience and a preliminary test at a higher SA concentration (0.5 mM), showed a slight reduction in germination under non optimal conditions (limited water availability and temperature), this prompted me to select a lower concentration for this study (0.1 mM), which is within the range of concentration tested in numerous studies of different crop species. I agree that a full dose response curve experiment would have strengthen this study, but, due to time constraints, it could not be performed (see later comment for more explanation). 

L 140 – imbibition duration - Similar to the apparent lack of a SA dose-response assessment, how do you know this length of soaking time was suitable/optimum? No preliminary study was performed to ensure that this time was optimal. 24 hours-time was chosen based on imbibition treatments previously reported in the literature on grass species. The rationale for this decision of 24 hours is to allow enough time for the seed to absorb water and SA, and interrupt imbibition before the germination process is irreversible. A preliminary study (whose results were confirmed in this study) determined that the earliest germination event was recorded between 48 and 72 hours across all three species making 24 hours a safe time to avoid germination.

Added sentence: “ , following previously described methodology for SA delivery to seeds [34,49]”.

L 145-146 – “dusted onto the rotating mass” - on Lines 143-144 you stated that the filler material was dusted on using a paint brush. But here you state that the powder was dusted on while seeds were rotating. This appears near impossible as you would disrupt the seed flow in the coater. Please clarify. This point was raised by both reviewers. I have added the following sentence to clarify the use of the paint brush in delivering the powder during the coating process: “Talc was used as the filler material. Cleaned seeds (10 g) were placed inside the rotary coater, with rotor speed set at 300 RPM, and seeds were initially exposed to liquid spray until moist before powder was dusted onto the rotating seed mass using a paint brush. By gently tapping the brush on the drum, the powder was slowly released on the rotating mass of seed without affecting their flow. “

L 148 – “multiple seeds” - unclear. Do you mean seeds clustered together? Not singular? Correct, I have rephrased the sentence: “Seeds were routinely checked to visually evaluate the even coverage of the coat, and to ensure singulation (i.e. each coated unit contains one seed)”

L 155 – “PEG solution was added weekly” - at each re-hydration point was the water potential checked to ensure your target water potential was maintained. This is critical because at some higher concentrations of PEG the water potential changes as water evaporates, and then at re-hydration with your target concentration this then leads to an altered water potential. No, the water potential was checked at the beginning of the experiment, but not weekly. Given that the germination experiment lasted for 3 weeks, I thought it was not necessary to check weekly for changes in water potential. But this is a valid point that I have not considered, that I would take in consideration in further experiments. However the variation in the water potential would have been similar across all the treatments , so for example, even though the water potential might have fluctuated between -1.2 and -1.4 Mpa, the same variation would have happened in all treatments.

L 158 - No mention of 0MPa for ‘control’ conditions. Added: In order to test whether SA improved germination success under water-limited conditions PEG 8000 (Sigma-Aldrich, St Louis, USA) diluted in deionised water at 0 (for control), 24.72, 30.78, and 35.90 g/l was used to obtain solutions of 0.0, -0.6, -0.9, and -1.2 MPa water potential at 20° C.

L 165 – “Non-viable seeds were excluded from the total” – Does this mean all data presented in the lab trials are cut-test adjusted? If you did do this, you need to clearly state and show these results for readers to interpret. For example, what if your SA imbibing treatments (or encrusting) lead to a significant drop in viability but you only present the viable component in graphs. This could mislead the reading to more positive findings. On average there was a 15-20% mortality due mostly to fungal/bacterial infection. I decided not to surface sterilise the seed prior testing germination because the treatment would not have been applicable to encrusted seed. This has resulted in some contamination. The rate of contamination was usually spread evenly across treatments and there was no clear trend of higher mortality for any specific treatment. I have provided supplemental information with the mortality and germination not adjusted for viability in appendix 3

L 167 – site selection - Is it a grassland requiring restoration? The whole Intro was pitched around grassland restoration. It would aid the reader here to provide some more background to the field trial system. L 169 - These mining operations from aerial image searches look like forested systems? If your focus is on grasslands please elaborate this mining example. The field trial is not on a restoration site, and it is not necessarily focused on grassland restoration. The part in the introduction about grassland has been removed. The site was chosen for availability and proximity to an area to be restored in the future. This trial is not directly involved with the restoration of that specific mine site, but the studied species might be used for restoration of the mine site as they naturally occur in the area. However, the main goal of the trial was to test the feasibility and effect of seed treatments on survival, not to simulate a “seeding for restoration” scenario.

L 173 – “the five treatments” - Which five. Re-iterate. Done

L 175 - It is unclear here whether your "replicates" here are the same as blocks in the following Line. If they are, the field layout appears to be pseudo-replicated where you have only tested your treatment in one large block containing 4 replicates. It is not replicated at the site level. How have you accounted for this in your analysis. Only testing in one site would potentially confound your findings. I have provided in the appendices a layout of the experimental plot (appendix 1). There was not enough space for replication at site level due to limited space enclosed by a fence. (117 m2). 

L 180 and 189 – related to the above comment it appears that your germination bags and plot experiment where sown outside of the replicated line experiment design leading to three independent experimental sub-sets (germ bags , line experiment and plot experiment). This confounds your experimental finding and conclusions making it difficult to directly compare (see comments later surrounding your conclusion of the density/competition work). This is one flaw of this work that needs addressing. Yes, this interpretation is correct. When the experiments were originally designed, it was not my intention to directly compare those three experiments. For this reason, I have not performed any analysis between the experiments. I have modified the way those results are interpreted in the discussion to acknowledge this limitation. 

L 194 - Write 45 in words. You typically do not start sentences with numbers. Fixed

L 208 – “emergence” - This is confusing. All this is worded for germination then you have mentioned emergence here? Rephrased to avoid confusion: “where: b is slope curvature, gmax is final germination and T50 is germination speed, intended as time (days/weeks) required to reach half of the final germination. Parameter comparison on final germination and germination speed were then performed to assess differences among treatment (significance P <0.05). The same analysis was performed on seedling emergence data.”

L 209 - Related to the above comment how does this work for emergence. This section needs revision to accommodate both life stages you are addressing. To the reader as it stands it is not clear what you have done.

The analysis was performed on both germination and emergence. Removed emergence and added this sentence: The same analysis was performed on seedling emergence data.

L 215-216 - This is very unclear to the reader. I have drawn multiple diagrams noting each of your treatment and I still cannot work out what you mean here. Un-treated control = easy to work out. "Treated without SA" is this the "imbibed without SA" or the "coated without SA"? Same goes for the "treated with SA". I would request a complete re-write so that we can critically review your results/figures/tables. Treatments were renamed across the result section to improve clarity and consistency. When treatment is grouped, they are presented as IS+ES or IN+EN. “To assess the effect of SA, seeds that were provided SA (via imbibition and encrusting, IS+ES) were compared to seeds that received the treatments without SA (IN+EN). If a significant difference was detected, SA delivery methods of encrusting (ES) and imbibing (IS) were then compared.”

L 232 – 243 - Grammar is very poor. Suggest a re-work. This sentence is not necessary, and I decided to remove it

L 236 – 237 Is both Encr and Imb here without salicylic acid (SA). For now, this is not very clear and finding it hard to work out what treatment you are comparing to what. And discussed later on, your treatment naming convention needs further thought. It is very hard to keep track of what treatment is what. Here you have Encr and Imb, later on you have SA and No. In the methods section I have provided definitions and acronyms for each of the 5 treatments and explained treatment grouping for analysis, and tried to use the abbreviation consistently thought the document.

L 239 - P value was lower case in stats section. Be consistent. All corrected to P

L 242 - More frequent measures of emergence may have provided some less variable results here. There is a very high range of variation between the 1 week and 2 week monitoring points which is a shame as this could have been powerful. When the study was initially designed, I thought that weekly scoring would have been enough, but I agree that higher frequency of scoring during the first 3 weeks would have been very useful. I would keep this in mind for further experiments. 

L 246 - It is unclear here which significance level this is related to? Assume it is for the Encr v the Ctrl? Not this latter, minor 4% difference? Added compared to ctrl

L 248 - Check wording here between the brackets. Some info for Ctrl is missing. Fixed

L 252 – 254 - This is best suited to the methods section. This also relates to my query on Line 215 – 216 about knowing what you are comparing. Unclear treatment names also add to this difficulty. I think this sentence is needed here to explain how the results are presented. Given the high number of combinations, (TRT*SPECIES*WATER POTENTIALS) I decided to group the treatments based on the presence of SA. One group is IN + EN, the other IS + ES. If there is a significant difference between the groups, then I compare the single treatments IS vs ES. 

L 255 - This is a discussion point. Sentence removed

L 252 – 256 - Now that I have read on a little more into your results here it is very unclear what treatment you are comparing to what.

As I read it you are comparing untreated (Ctrl) seeds to (1) seeds that received SA via imbibition, but not encrusted ("NO") and (2) seeds that DID NOT receive SA via imbibition, but were received SA via encrusting ("SA" in your figures")

If this is true, then this small sub-trial work is unbalanced and slightly confounded. You are missing a treatment of untreated seeds that are encrusted without SA imbibition and seeds that were imbibed with SA and encrusted. One example of the confounding factor here is that one could argue that just encrusting seeds alone could provide improved water relations for seed germination at lower water potentials. As you do not have this treatment comparison to compare all findings are speculative. This is not the case, and the original wording was confusing. Now that treatments have been renamed hopefully it will be clearer. What in the original version was named SA, combines the results of IS (imbibed with SA) and ES (Encrusted with SA), whilst NO combines the results of IS (imbibed with SA) and ES (Encrusted with SA). All treatments are taken in consideration. 

Line 257 - What is this statistical result referring to. If it is for Cntrl v SA then according to your post-hoc lettering above the germination columns at 0 MPa this is not true. They both share a small lower case "b". Recommend checking your numbers. Imbibed in water, if this is what 'NO' is (very hard to tell!), then there appears to be a significant increase in water imbibition when compared to SA imbition

It's not compared to the control, but with IN+EN. The sentence was changed as follow. “In A. scabra final germination of SA treated seeds (IS + ES), at 0.0 MPa water potential, was 4.3% lower than seeds treated without SA (IN + EN) (P < 0.05) (Fig 4)”

L 258 - Quality control on what has been submitted? Assume this is for figure 4? In the originally submitted file, the cross link between figure caption and in text reference was working correctly. Something has gone wrong during or after submission. To avoid broken links I will not use this functionality going forward.

L 261 – wording in brackets - What is this being compared to? There is not mention here of the Ctrl. The comparison is between (IN + EN) vs (IS + ES)

L 262 - Many in-text figure citations are missing making it hard to link the results to the figures. Added reference to figure throughout the result section. 

L 263 – I suggest to not use the word optimal for your MPa conditions. This is an assumption that non water limited conditions is optimal. Replaced with “full water availability” or “0.0Mpa”

L 264 – ‘SA delivered through encrusting…’ - Example of confounded data here. What would happen if you just encrusted the untreated seed? Data in Figure 3 (your previous work suggest the M stipoides benefits from encrusting alone - ca. 80%+ germination). But you do not have this comparison here under your water potential experiment here to conclude how it would perform. The encrusting of untreated seeds is represented by EN.

L 285 - Which Figure am I looking at now? Assuming 5C but I should have to work it out. I’ve rearranged the images in fig 5 to match the description and add fig references to the text. 

L 286 – compared to seeds treated without SA - so are you referring to your "NO" treatments. This is very hard to follow. Recommend a major updated to your treatment naming convention. If your methods you just word that you have five treatments. Then at times we have "Ctrl", "No", "SA" and even "ES" and "IS" on Line 254 than I still haven't seen. Naming convention updated across the document

See previous comments. The new naming convention is now used in this sentence “Plants emerging from SA treated seed (IS + ES), compared to seeds treated without SA (IN + EN)…”

L 335-337 – “A potential explanation….(petri dish).” - This is highly speculative and an odd discussion point. First, soaking seeds for 24 hours in either water or water containing beneficial agents is a well-established technique for use in native seeds. Even some of the authorship group here have published many papers in this space. So, I find the suggestion of anoxic stress during 24h of soaking to be off track. Further there is no measured evidence provided in this study to suggest this could be present. Second, the suggestion of anoxic environments due to extended time spent in a petri dish is even more out there. International seed testing standards/protocols for germination testing in both agriculture and native seeds use Petri dish work to carry out these standard assessments and have been standard practice for centuries. For instance, the classic Baskin and Baskin book reports germination data for 15,000+ species largely from Petri dish work. ISTA seed testing guidelines recommend the same approach. Even the authorship group have just set some standards in this very space (Pedrini and Dixon 2020, Rest Ecol, "Guidance Statement 13.2:). I find this a nonsensical discussion point and it should be removed or heavily backed up by evidence using native seeds (most evidence reported below this point is agricultural).

L 338 – oxygen availability - Same as previous comment. All speculation.

L 341 – 342 - Same as previous comment. All speculation.

Section removed. In the originally submitted manuscript the discussion was trying to focus on every different treatment response in every scenario (e.g. reduced water availability, field emergence) and provide justification for each variation, resulting in lots of unsubstantiated speculation. I decided to merge the three sections and focus the discussion on the more relevant trends, limiting the interpretation of the results to what was observed. I think this approach improves the flow of the manuscript and provide a stronger focus on the most meaningful trends. (germination and emergence are not affected by SA application, but survival and growth are).

L 352-353 - One key step that has been missed in this study is some detailed preliminary work on the concentrations of SA required and duration of soaking needed to test whether there will be a benefit at all for this use in native seeds. The authors clearly stated "However, it has never been tested on native species for ecological restoration", Line 94-95, yet they jumped into testing one concentration and one soaking time that worked on agricultural seeds. Then most of this discussion hovers around maybe the concentration could be altered. This study would have been strengthened by detailed dose-response research into the most suitable SA treatment and soaking duration.

I agree with this statement that a dose response curve would have been very informative. unfortunately, due to time constraints, such a preliminary experiment could not be set up. However, the fact that SA benefits were detected consistently in plant survival suggests that this concentration was appropriate. Statement about SA concentration in the discussion have been removed. 

L 366-368 – “SA also provided….without interspecific competition” - As discussed previously, this finding would also be confounded by the experimental design of this study. It severely limits the interpretation provided here. For instance, the discussion point here is saying that SA provided a significant improvement in plant survival in both competition scenarios, yet, without a fully balanced treatment design the authors cannot outright conclude that SA provided all this benefit. What would happen is just encrusted (no SA) were tested? Without it these results and conclusions are flawed. See above comments on treatment grouping for analysis. Hopefully, the updated treatment naming and grouping would make this point clear. I removed any mention of competition and replaced it with the name of the experiments (line and box) to keep consistency across the paper.

L 376-379 - Once again, speculative and not backed up by data. From an experimental design perspective, the low and high density are not directly comparable. They are independent assessments that were set-up very differently. The 'line experiment' was done in 1m long strips by 5cm wide, sowing 100 seeds (roughly equating to one seed per 5cm2) - presumably in the replicated plot design outlined on lines 173-176. Whereas, the 'plot experiment' was done in 50 x 50cm2 plots, sowing of 100 seeds over this 2500cm2 (1 seed per 25cm2) seeded in plots outside of the line experiment area. So straight up, the plot design (line v plot) and thinning protocols are different (e.g. the plot experiment was thinned at 4 weeks, but nothing was done at the line experiment). Further, as the seeded areas were in different locations, site level variation could have driven these differences observed. Replicate numbers for the 'plot' study are not mentioned anywhere that is obvious. 4 replicates It is now added in the methods section To be truly comparable it would have been best to have this assessment carried out with the same sowing technique - either as line or plots - with known differences in seed sown at Day 0 or thinned at a point down to target densities. This would make a direct comparison possible. As it stands, again the design in problematic. This limitation is now acknowledged in the manuscript: “Even though these two experiments are not directly comparable, the differences in the two competition scenarios might provide useful insight to restoration practitioners. … This factor needs to be taken in consideration when planning for seeding operation, and further studies designed to directly compare the effect of seeding density on plant survival could provide restoration practitioners with more accurate seeding rate prescriptions that could help limit wastage of valuable and expensive seeds [58]”

L 392- 393 - Was the Hobo logger calibrated for the soil type of the field site? Where is this data presented? Where any soil analyses completed? I have not performed a soil analysis but have tested the water retention curve for this soil collected from the filed site to relate water volumetric content in soil to water potential (Mpa). See in attached supplemental material annex 1 (SoilWaterRetentionCurve.pdf)

L 399-400 - See discussion points mentioned previously aligned to the unbalanced treatment design and confounding concerns. Not sure if this SA significant effect can be set in stone. See above comments on treatment grouping for analysis. Hopefully, the updated treatment naming and grouping would make this point clear.

L 403-404 - How can you state that there was a summer drought when the site got close to 210mm over the summer period! Rephrased: “dry summer months.” Even though 210 mm seems much, most of the rain dropped in a few hours during two storms. 

L 455 – many reference need checking for complete information or format. E.g. ref 14 is LOUD. Fixed

How could I find ref 38? https://www.dpi.nsw.gov.au/agriculture/pastures-and-rangelands/rangelands/publications-and-information/grassedup added to citation

Check out ref 49: Updated and corrected

Figure 1 – Austrostipa and Rytidosperma both not spelled correctly. Corrected

Figure 4 – check spelling again for Austro Corrected

Figure 6 – A. stipa????????? Corrected

Supp Material file - species spelling again fixed

 

Reviewer #2: Pedrini et al. research provides a valuable contribution to the field of restoration ecology by demonstrating that salicylic acid can be applied within a seed coating to improve drought tolerance of important grass species in Australia. This work also has implications for improving seeding efforts in dryland systems across the globe.

The authors need to improve the clarity in their methods section. For example, I would like to see a better description on why the different studies were performed and how they related to their hypotheses. In the experimental design section, I’ve added a table that shows how hypotheses and experiments are related. 

Increased detail and accuracy could also be given in describing the seed treatments used in the different studies and how the results were analysed. Description of seed treatment in the methods section was improved.

They state that five treatments were tested in all studies, but the field study results only have three seed treatments. The number of treatments was still five, but when presenting the results, the treatments were grouped (see previous responses)

I had a number of suggestions throughout the manuscript to improve grammar, clarity, and presentation of the study (see attached pdf). My edits were not exhaustive but show examples of issues that should be considered. The authors also need to write a conclusion. After some work this manuscript will be a quality publication. Most of the edits suggested by the reviewers have been included in the manuscript. A selection of point raised by the reviewer in the PDF file have been reported here. 

Line 65 consider adding a citation here for this point - added citation: Brown SL, Reid N, Reid J, Smith R, Wal Whalley RDB, Carr D (2017) Topsoil removal and carbon addition for weed control and native grass recruitment in a temperate-derived grassland in northern New South Wales. Rangeland Journal 39:355–361

Line 99 Consider adding more justificaiton in the introduction for testing SA delivery through coating or imbibition – “Moreover, by comparing SA delivery methods of imbibition and coating, we test the efficiency and viability of seed coating technology in delivering the benefits of SA to native grass seeds.”

Line 103 There is nothing in the introduction that justifies this portion of the study, i.e. seeding at different rates. After reading over the paper I see that your study design was not performed to test SA influence in seeding success with high an low intraspecific competition. You had two different studies with different plant densities. It is fine to make inferences from the results of these studies in the discussion section but this is not a testable hypothesis and I remove this part from the sentence

It's a good point. This part was removed. 

Line 144 Did you really use a brush? If not consider taking out the last part of this sentence after the word "seeds" - yes, i used a paint brush. When dealing with small amount of seed, especially in the first step of seed coating, gently tapping on the paint brush allows for a gentle and even distribution of powder over the rotating seed mass.

Line 176 need to define and justify what these studies are: The method sections was restructured to better explain the experiment and their role in addressing the three hypotheses.

Line 187 It is not clear from the previous text why you are doing research under low intraspecfic competition. I would add something to the start of this experiement to justify this work. I assume from the previous experiment you had different levels of competition due to a treatment effect from SA. Subsequently, an additional study was implemented to better quantify SA impact on plant survival under similar levels of intraspecific competition. 

All reference to high and low intraspecific competition were removed from the method section. The justification for thinning in the plot experiment is explained as follow: “Thinning was performed to limit potential intraspecific competition that might have altered plant growth. “ 

Line 232 this sentence is difficult to read, please expand with additional sentences to improve clarity. This sentence is not necessary, and I decided to remove it

Line 279 grammar a little rough. It would also be useful if you used the name of the experiments defined previously in the headings of the methods. This type of error seems to be an issue throughout the manuscript. Name of experiments has now been used consistently throughout the document. 

Line 298 as I read the methods it appeared their were multiple treatments in the study. Here you are only reporting two treatments and it is not clear what the SA treatment is. Please try to improve clarity in the methods section on what treatments are being tested. You also need to write separate statistical analysis for each of the different studies due to their different treatments. See previous reply to reviewer 1. All treatments were tested but in presenting the data, the treatments were group. Hopefully, the new naming standards makes this approach clearer.

Line 436 Not critical but as a reader I like to hear the authors discuss future work that should be done to adopt the technologies and where the technology could potentially be applied. For example, could and SA coating improve resistance to freezing, pathogen, or other types of stress. - PLOS 1 papers have conclusions and this paper is missing this section. Add a conclusion: This study showed that seed coating with salicylic acid can improve survival of three Australian native grass species used in restoration programs. However, further research is needed to evaluate SA coating capability to improve native plant resistance to different biotic and abiotic stresses such as extreme temperatures, salinity, pathogens, and herbicides. Moreover, coating with SA in combination with other beneficial compounds should be tested on restoration priority species, because their combined impact on seed germination, emergence, growth and plant establishment could improve the successful deployment of native seed onto degraded landscapes, ultimately allowing for a more cost-effective seed-based restoration

---

## [Decision Letter · Decision Letter 1]

15 Mar 2021

PONE-D-20-33303R1

Seed encrusting with salicylic acid: a novel approach to improve establishment of grass species in ecological restoration

PLOS ONE

Dear Dr. Pedrini,

Thank you for submitting your manuscript to PLOS ONE. After careful consideration, we feel that it has merit but does not fully meet PLOS ONE’s publication criteria as it currently stands. Therefore, we invite you to submit a revised version of the manuscript that addresses the points raised during the review process.

Although the reviewers of your manuscript came to very different conclusions regarding its readiness for publication, I have carefully read the manuscript myself and found that I concurred with the assessment provided by Reviewer #2. 

We look forward to receiving your revised manuscript.

Kind regards,

Craig Eliot Coleman, PhD

Academic Editor

PLOS ONE

Journal Requirements:

Reviewers' comments:

Reviewer's Responses to Questions

**Comments to the Author**

1. If the authors have adequately addressed your comments raised in a previous round of review and you feel that this manuscript is now acceptable for publication, you may indicate that here to bypass the “Comments to the Author” section, enter your conflict of interest statement in the “Confidential to Editor” section, and submit your "Accept" recommendation.

Reviewer #1: (No Response)

Reviewer #2: All comments have been addressed

2. Is the manuscript technically sound, and do the data support the conclusions?

Reviewer #1: No

Reviewer #2: Yes

3. Has the statistical analysis been performed appropriately and rigorously? 

Reviewer #1: No

Reviewer #2: Yes

4. Have the authors made all data underlying the findings in their manuscript fully available?

Reviewer #1: Yes

Reviewer #2: Yes

5. Is the manuscript presented in an intelligible fashion and written in standard English?

Reviewer #1: Yes

Reviewer #2: Yes

6. Review Comments to the Author

Reviewer #1: Pedrini et al have provided a revised version of their manuscript based off back from two reviewers. One of these reviews was provided by me.

After re-reading the manuscript I still do not feel the two major issues have been addressed suitably for progressing this paper to publication.

First, even though the authors have provided a map of the field design (“S1_FieldExperimentLayout) and some adjustment to the description in the methods and review responses (“limited space that was fenced”), the authors have not addressed the pseudo-replication of this study. The “Line”, “Bags”, and “Box” treatment appear to be replicated at least 4 x times within the trial area, however, this is not replicated at the site level. This lack of site level replication needs to be addressed as it is a flaw in the experimental design that could influence the findings and overall conclusions.

Second, and after the authors attempted to clarify the use of treatments and the use of acronyms, it is now apparent that the way treatments were ‘lumped’ and statistically analysed is problematic. For instance, the authors lump salicylic acid (both soaked “IS” and encrusted “ES” = IS+ES) treatments together along with the non-salicylic acid treatment (IN+EN) and carry out one level of analysis compared against un-treated control seeds. These two treatments within each lumping are not independent and I fail to see how you can validly lump them together for analysis and then separate them again later for a second post hoc assessment as independent treatments (if significance was found in the first step). Further, at the very basic level of tested seed treatments it is very odd to group something like seeds that have been soaked in SA with seeds that have been encrusted with SA – they are two different approaches and it makes no scientific sense to group them. Extended commentary on this flaw is provided below but a fear there is a lack of treatment independence that makes any findings or conclusions of this study inaccurate. Until this data is presented and analysed in a different manner, I do not feel this paper is suitable for publication.

Line by line comments provided below for other minor comments/thoughts (note, no detailed review was provided from the discussion down as the methods/results must be fixed first):

L 32-33 - would suggest swapping seedling growth to before plant survival

L 33 - Suggest spell out SA at the start of a sentence

L 41 - where are the figures to support the ‘cost-effective’ nature of this comment.

L 60 – Not sure if this sentence quite works in the way it is presented. Seed deployment is highly efficient in many degraded systems. For instance, native seed is spread very efficiently across the western USA using traditional sowing practices such as broad acre drilling. So, I would argue that its not ‘deployment’ but more so the conversion of this seed to an established plant that needs improvement in efficiency (e.g. "efficiency of seed deployment" is not quite the problem)

L 72 - Unusual for acronyms to be used to start a sentence just like numerals. Suggest to spell out throughout paper..

Line 110 – Use of caryopsis appears incorrect. From the wording I would assume this would be the florets being immersed. Please clarify.

L 134 – “in a 15cm”

L 160 - desiccation of what?

L 161 - sufficient hydration of what?

L 164 - The use of 'this value' here is unclear as is infers ‘one value’ and then a range is referred to for water available - please clarify language.

Do you mean "this lab-based water potential gradient" resembles the range of water available recorded in the field during the winter months. If this is what is meant, this is not reflected in the field data as volumetric water content appears to not exceed 0.17 or 17%. Also the field data over your winter months doesn’t seem to represents anything close to 'field capacity’ with a rough conversion of of this data to be >0.2MPA. So, the usefulness of lab comparisons to field results would be questionable.

L 216-217 - These are not independent treatments and I cannot not see how this is a valid lumping of independent variables. See comments further down.

L 249 (whole section) - From my first review I appreciate that the authors have attempted to clarify the use of the treatment acronyms in the worded text and figures. In this revised version I now worry that the study design and the way it is presented is problematic.

It now appears, that at least in this section, the treatments have been lumped together in the data presentation and statistical assessment.

This is a major problem for the paper. Unless there is a new way to lump two "independent" treatments together (e.g. in this instance IS & ES 'lumped' and IN & EN 'lumped') I cannot see how you can conclude the success of any of these treatment over another.

For what I can tell from your statistics methods (ca. Line 216) you have analyzed SA 'lumped' together and the two non-SA treatments 'lumped' together against the un-treated CTRL. Then if statistically significant effects were found you then assess the ‘within’ SA or non-SA comparisons (here for instance whether ES was different to IS). The first lumping is just not independent (i.e. 4 treatments lumped into 2 groups) and I struggle to see how this a valid approach to take the next ‘within’ post hoc step (compare single independent treatment informed from the first lumping).

To me this section needs a solid re-write as there are two independent lines of investigation that needs to be presented separately or in a different manner.

In one set there could be a clear presentation of CTRL v IN v IS and in the second CTRL v EN v ES. If SA provides the hypothesized desired benefits, then you can conclude its treatment benefit over the CTRL and whether the SA treatment worked over the soaking in water or coating with no-SA.

An alternative is to provide all treatments in the column graphs of Figure 4 (e.g. five columns with CTRL, IN IS EN ES) and compare each of these independent treatments (IN IS EN ES) back to the untreated control (CTRL). Using something like a post hoc pairwise comparison such as Dunnett (using the ghlt function in the multcomp package in R which deals with many treatments being compared against one control) you can them map the effect size of these independent treatments against the focal control, and even to each other, to offer a valid independent conclusion.

As it stands, I think this analysis and data presentation is a problem and a major flaw.

L 281 (whole section) - My commentary is repeated here from the above comments surrounding the lumping and lack of Independence.

Reviewer #2: In general my comments have been addressed by the authors and I feel the manuscript will be ready for publication after some minor editing. There is still some grammatical issues in the manuscript. Some of examples are listed below

line 102, this phase is not clear "and utility as pasture."

line 174, "at 1cm depth" change to "at a 1 cm depth"

Line 241, need a period. Also on this line and through the manuscript you may want to add the word "the" before saying the treatment name. For example on this line it says ..."IN seeds showed no significant difference compared to CTRL." Perhaps you could say "IN seeds showed no significant difference compared to the CTRL." Additionally, in my opinion the use of the word "significant" is redundant. My preference would be to have the sentence say "IN seeds showed no difference compared to the CTRL." With that said, this is a style preference and the authors do not need to implement this change here and at other parts of the manuscript.

Line 298, poor grammar

Line 303, for ease of reading would it may be better to keep the treatment order the same as the sentence above

Line 309, Need period

Lines 396 - 401, very long sentence with poor grammar

Lines 397 - 401, consider expanding your discussion of the James et al. (2011) paper to talk about how your technology has application outside of the region this study was conducted. James et al. (2011) implemented a dormant seeding with seeds remaining in the soil under freezing soil conditions for 4-5 months prior to emergence. In cold desert regions similar to where James et al. (2011) conducted their study, high mortality can occur over the winter to the seeds and unemerged seedlings from freezing temperatures, pathogens, drought, and other stressors. Just as the SA treatment improved drought stress in your study it is probable that a SA seed treatment may mitigate limiting abiotic conditions associated with dormant winter seedings.

If you chose to add this topic to your discussion I would also cite James et al. (2019).

James, J. J., R. L. Sheley, E. A. Leger, P. B. Adler, S. P. Hardegree, E. S. Gornish, and M. J. Rinella. 2019. Increased soil temperature and decreased precipitation during early life stages constrain grass seedling recruitment in cold desert restoration. Journal of Applied Ecology 56:2609-2619.

7. PLOS authors have the option to publish the peer review history of their article (what does this mean?). If published, this will include your full peer review and any attached files.

Reviewer #1: No

Reviewer #2: **Yes: **Matthew Madsen

---

## [Author Response · Author response to Decision Letter 1]

29 Apr 2021

Dear editor and reviewers

Thank you for providing further review and feedback on the manuscript. We have addressed the reviewers’ comments, noting the editor’s support for the opinion of Reviewer 2. I hope that this response and revision of the manuscript satisfactorily address the reviewers’ concerns. 

Kind regards

Simone Pedrini

Review Comments to the Author

Reviewer #1: Pedrini et al have provided a revised version of their manuscript based off back from two reviewers. One of these reviews was provided by me.

After re-reading the manuscript I still do not feel the two major issues have been addressed suitably for progressing this paper to publication.

First, even though the authors have provided a map of the field design (“S1_FieldExperimentLayout) and some adjustment to the description in the methods and review responses (“limited space that was fenced”), the authors have not addressed the pseudo-replication of this study. The “Line”, “Bags”, and “Box” treatment appear to be replicated at least 4 x times within the trial area, however, this is not replicated at the site level. This lack of site level replication needs to be addressed as it is a flaw in the experimental design that could influence the findings and overall conclusions. 

What is to be consider pseudo-replication is still a controversial issue with useful commentary relevant to the biological sciences outlined in Davies GM, Gray A (2015) Don’t let spurious accusations of pseudo-replication limit our ability to learn from natural experiments (and other messy kinds of ecological monitoring). Ecology and Evolution 5:5295–5304. The trial area in the study site is a unique region within a high-profile national park where the study species naturally occur. As outlined in Davies and Gray (2015) there are particular reasons when it is permissible to conduct experiments within one site location when constraints such as ‘avoidance of clearing other areas in a national park’ are important concerns. The trials were replicated albeit within a fenced area 

Second, and after the authors attempted to clarify the use of treatments and the use of acronyms, it is now apparent that the way treatments were ‘lumped’ and statistically analysed is problematic. For instance, the authors lump salicylic acid (both soaked “IS” and encrusted “ES” = IS+ES) treatments together along with the non-salicylic acid treatment (IN+EN) and carry out one level of analysis compared against un-treated control seeds. These two treatments within each lumping are not independent and I fail to see how you can validly lump them together for analysis and then separate them again later for a second post hoc assessment as independent treatments (if significance was found in the first step). Further, at the very basic level of tested seed treatments it is very odd to group something like seeds that have been soaked in SA with seeds that have been encrusted with SA – they are two different approaches and it makes no scientific sense to group them. Extended commentary on this flaw is provided below but a fear there is a lack of treatment independence that makes any findings or conclusions of this study inaccurate. Until this data is presented and analysed in a different manner, I do not feel this paper is suitable for publication. 

This comment is addressed as below. 

Line by line comments provided below for other minor comments/thoughts (note, no detailed review was provided from the discussion down as the methods/results must be fixed first):

L 32-33 - would suggest swapping seedling growth to before plant survival .

Done

L 33 - Suggest spell out SA at the start of a sentence. 

Done

L 41 - where are the figures to support the ‘cost-effective’ nature of this comment. 

Good point. A cost benefit analysis was not performed in this study. The sentence has been rephrased as follow: Overall, adjustment of seeding rate according to expected emergence combined with the use of salicylic acid via coating could improve seed use efficiency in seed-based restoration.

L 60 – Not sure if this sentence quite works in the way it is presented. Seed deployment is highly efficient in many degraded systems. For instance, native seed is spread very efficiently across the western USA using traditional sowing practices such as broad acre drilling. So, I would argue that its not ‘deployment’ but more so the conversion of this seed to an established plant that needs improvement in efficiency (e.g. "efficiency of seed deployment" is not quite the problem). 

My experience in Europe and Australia is that delivery is still a major bottleneck, especially when dealing with diverse seed mixes with complex morphological seed structures (such as the grasses in this study). However, I agree that seed deployment alone is not the main issue. I have therefore rephrased the sentence as follows: Given the high cost and often highly limited availability of native seed, improving the efficiency of seed use (deployment, germinaition, and plant establishment) is crucial if ecological restoration is to be delivered at the expected landscape scale.

L 72 - Unusual for acronyms to be used to start a sentence just like numerals. Suggest to spell out throughout paper. 

Done

Line 110 – Use of caryopsis appears incorrect. From the wording I would assume this would be the florets being immersed. 

Yes, caryopsis in line 113 was incorrect, and has been replaced with floret: …with complete immersion of the floret in a 50% sulphuric acid solution…

L 134 – “in a 15cm” 

Done

L 160 - desiccation of what? Filter paper, 

added

L 161 - sufficient hydration of what? 

Filter paper, added. Germination tests were performed in Petri dishes lined with two filter papers moistened with 14 ml water or Polyethylene Glycol (PEG) solution and placed in sealed plastic bags to reduce filter paper desiccation. Each week, 2 ml of water or PEG solution was added to ensure sufficent hydration of the filter paper. 

L 164 - The use of 'this value' here is unclear as is infers ‘one value’ and then a range is referred to for water available - please clarify language.

Do you mean "this lab-based water potential gradient" resembles the range of water available recorded in the field during the winter months. If this is what is meant, this is not reflected in the field data as volumetric water content appears to not exceed 0.17 or 17%. Also the field data over your winter months doesn’t seem to represents anything close to 'field capacity’ with a rough conversion of of this data to be >0.2MPA. So, the usefulness of, ab comparisons to field results would be questionable. 

I agree with this point. The field site never reached field capacity. I have rephrased the sentence to acknowledge this: The values at reduced water potential resembles the range of water availability recorded in the field during the winter months. (S2 Appendix, Soil Water retention curve).

L 216-217 - These are not independent treatments and I cannot not see how this is a valid lumping of independent variables. See comments further down. L 249 (whole section) - From my first review I appreciate that the authors have attempted to clarify the use of the treatment acronyms in the worded text and figures. In this revised version I now worry that the study design and the way it is presented is problematic. It now appears, that at least in this section, the treatments have been lumped together in the data presentation and statistical assessment. This is a major problem for the paper. Unless there is a new way to lump two "independent" treatments together (e.g. in this instance IS & ES 'lumped' and IN & EN 'lumped') I cannot see how you can conclude the success of any of these treatment over another. For what I can tell from your statistics methods (ca. Line 216) you have analyzed SA 'lumped' together and the two non-SA treatments 'lumped' together against the un-treated CTRL. Then if statistically significant effects were found you then assess the ‘within’ SA or non-SA comparisons (here for instance whether ES was different to IS). The first lumping is just not independent (i.e. 4 treatments lumped into 2 groups) and I struggle to see how this a valid approach to take the next ‘within’ post hoc step (compare single independent treatment informed from the first lumping). To me this section needs a solid re-write as there are two independent lines of investigation that needs to be presented separately or in a different manner. In one set there could be a clear presentation of CTRL v IN v IS and in the second CTRL v EN v ES. If SA provides the hypothesized desired benefits, then you can conclude its treatment benefit over the CTRL and whether the SA treatment worked over the soaking in water or coating with no-SA. An alternative is to provide all treatments in the column graphs of Figure 4 (e.g. five columns with CTRL, IN IS EN ES) and compare each of these independent treatments (IN IS EN ES) back to the untreated control (CTRL). Using something like a post hoc pairwise comparison such as Dunnett (using the ghlt function in the multcomp package in R which deals with many treatments being compared against one control) you can them map the effect size of these independent treatments against the focal control, and even to each other, to offer a valid independent conclusion. As it stands, I think this analysis and data presentation is a problem and a major flaw. L 281 (whole section) - My commentary is repeated here from the above comments surrounding the lumping and lack of Independence.

Answer: I understand the concern of the reviewer on the approach of lumping different treatments. When I was originally analysing the data, I have tested similar approaches to the one suggested by the reviewer by comparing each treatment to each other. However, this approach made the interpretation of the results harder and more confusing and did not allow direct testing of the hypotheses. The way the current analysis is performed, and the results presented, allows each hypothesis to be sequentially addressed (1. is SA beneficial? 2. Which method of delivering SA works better?). To address the reviewer’s concern on the validity of the results, I have provided bar chart graphs, with all five treatments (Ctrl, IN, IS, EN, ES) presented side by side, for all lab germination and field emergence (S4_GerminationEmergenceAnalysis.pdf) and bag germination, in line and box survival, and growth (S5_FieldExperimentsResults.pdf.) Those graphs confirm the same trends that are presented with the current analysis: SA does not improve germination and emergence, but provides benefits in terms of survival and, to a lesser extent, growth. As this concern was not raised by the other reviewer and the editor, I would prefer keeping the analysis, results, and discussion as they currently are. If the editor or the other reviewer think it is better to change the statistical approach and rewrite the result and discussion accordingly, I will do so but it would not materially affect the outcomes with both approaches being valid.

Reviewer #2: In general my comments have been addressed by the authors and I feel the manuscript will be ready for publication after some minor editing. There is still some grammatical issues in the manuscript. Some of examples are listed below

line 102, this phase is not clear "and utility as pasture." 

rephrased: “were selected on the basis of their predominance in revegetation and restoration activities and their potential use for pasture in replacement of, mostly European, fodder crop.

line 174, "at 1cm depth" change to "at a 1 cm depth" 

fixed

Line 241, need a period. Also on this line and through the manuscript you may want to add the word "the" before saying the treatment name. For example on this line it says ..."IN seeds showed no significant difference compared to CTRL." Perhaps you could say "IN seeds showed no significant difference compared to the CTRL." Additionally, in my opinion the use of the word "significant" is redundant. My preference would be to have the sentence say "IN seeds showed no difference compared to the CTRL." With that said, this is a style preference and the authors do not need to implement this change here and at other parts of the manuscript. 

Fixed here and throughout the document. The term significant was left when P value was provided and in cases that a difference was detected but was not statistically significant. E.G “SA treatments generally showed a slight but non-significant improvement in final germination”

Line 298, poor grammar 

Rephrased: To evaluate treatment effects on plant growth plant height and above ground dry biomass were recorded.

Line 303, for ease of reading would it may be better to keep the treatment order the same as the sentence above: 

sentence rephrased: Dry above-ground biomass was also higher in SA treatment (IS+ES, 3.4 g ± 0.22g) compared to non-SA treatments (IN+EN, 2.7 g ± 0.25 g) and untreated controls (CTRL, 2.2 g ± 0.25 g)

Line 309, Need period. 

Fixed

Lines 396 - 401, very long sentence with poor grammar 

Rephrased: “A study evaluating the demographic processes that influence established of native species from seed in field trials, performed by James et al. (2011), identified the main bottleneck to successful seedling recruitment as the emergence phase (when germinated seeds failed to express surface emergence )”

Lines 397 - 401, consider expanding your discussion of the James et al. (2011) paper to talk about how your technology has application outside of the region this study was conducted. James et al. (2011) implemented a dormant seeding with seeds remaining in the soil under freezing soil conditions for 4-5 months prior to emergence. In cold desert regions similar to where James et al. (2011) conducted their study, high mortality can occur over the winter to the seeds and unemerged seedlings from freezing temperatures, pathogens, drought, and other stressors. Just as the SA treatment improved drought stress in your study it is probable that a SA seed treatment may mitigate limiting abiotic conditions associated with dormant winter seedings.

This is a good discussion point and was added at the end of the section: The improved establishment of Australian grasses treated with SA suggest that a similar approach can be tested in different systems. For example, in cold desert restoration, where high seedling mortality occurs in the winter months due to a variety of biotic and abiotic stressors such as freezing temperature, drought and pathogens, [12,58] SA application to seed prior to seeding could be a useful adjuvant to promote seedling survival. 

If you chose to add this topic to your discussion I would also cite James et al. (2019).

James, J. J., R. L. Sheley, E. A. Leger, P. B. Adler, S. P. Hardegree, E. S. Gornish, and M. J. Rinella. 2019. Increased soil temperature and decreased precipitation during early life stages constrain grass seedling recruitment in cold desert restoration. Journal of Applied Ecology 56:2609-2619.

Citation added

---

## [Editor Report · Decision Letter 2]

5 May 2021

Seed encrusting with salicylic acid: a novel approach to improve establishment of grass species in ecological restoration

PONE-D-20-33303R2

Dear Dr. Pedrini,

We’re pleased to inform you that your manuscript has been judged scientifically suitable for publication and will be formally accepted for publication once it meets all outstanding technical requirements.

Kind regards,

Craig Eliot Coleman, PhD

Academic Editor

PLOS ONE
---

## [Editor Report · Acceptance letter]

18 May 2021

PONE-D-20-33303R2 

Seed encrusting with salicylic acid: a novel approach to improve establishment of grass species in ecological restoration 

Dear Dr. Pedrini:

I'm pleased to inform you that your manuscript has been deemed suitable for publication in PLOS ONE. Congratulations! Your manuscript is now with our production department. 

Kind regards, 

on behalf of

Dr. Craig Eliot Coleman 

Academic Editor

PLOS ONE